# Structure and function of the soil microbiome underlying $N_2O$ emissions from global wetlands

Mohammad Bahram [1,2,9✉], Mikk Espenberg [1,9], Jaan Pärn [1,9], Laura Lehtovirta-Morley[3], Sten Anslan [1], Kuno Kasak[1], Urmas Kõljalg[1], Jaan Liira [1], Martin Maddison [1], Mari Moora [1], Ülo Niinemets[4], Maarja Öpik[1], Meelis Pärtel [1], Kaido Soosaar [1], Martin Zobel [1], Falk Hildebrand[5,6], Leho Tedersoo[7,8,9] & Ülo Mander [1,9]

Wetland soils are the greatest source of nitrous oxide ($N_2O$), a critical greenhouse gas and ozone depleter released by microbes. Yet, microbial players and processes underlying the $N_2O$ emissions from wetland soils are poorly understood. Using in situ $N_2O$ measurements and by determining the structure and potential functional of microbial communities in 645 wetland soil samples globally, we examined the potential role of archaea, bacteria, and fungi in nitrogen (N) cycling and $N_2O$ emissions. We show that $N_2O$ emissions are higher in drained and warm wetland soils, and are correlated with functional diversity of microbes. We further provide evidence that despite their much lower abundance compared to bacteria, nitrifying archaeal abundance is a key factor explaining $N_2O$ emissions from wetland soils globally. Our data suggest that ongoing global warming and intensifying environmental change may boost archaeal nitrifiers, collectively transforming wetland soils to a greater source of $N_2O$.

[1] Institute of Ecology and Earth Sciences, University of Tartu, Tartu, Estonia. [2] Department of Ecology, Swedish University of Agricultural Sciences, Uppsala, Sweden. [3] School of Biological Sciences, University of East Anglia, Norwich, UK. [4] Institute of Agricultural & Environmental Sciences, Estonian University of Life Sciences, Tartu, Estonia. [5] Quadram Institute Bioscience, Norwich, Norfolk, UK. [6] Digital Biology, Earlham Institute, Norwich, Norfolk, UK. [7] College of Science, King Saud University, Riyadh, Saudi Arabia. [8] Mycology and Microbiology Center, University of Tartu, Tartu, Estonia. [9] These authors contributed equally: Mohammad Bahram, Mikk Espenberg, Jaan Pärn, Leho Tedersoo, Ülo Mander. ✉email: bahram@ut.ee

D espite covering only 8% of the terrestrial Earth surface, wetland soils (including gley and peat soils) store one of the largest organic carbon (C) stocks. Microbial degradation of C and nitrogen (N) stocks can lead to substantial releases of greenhouse gases (GHGs), including nitrous oxide ($N_2O$). $N_2O$ is a potent GHG with a global warming potential 265 times that of $CO_2$. $N_2O$ is the most important ozone-depleting substance[1]. This is particularly alarming as microbial sources of $N_2O$ may shift with environmental changes. Wetland soils are increasingly subject to land-use changes such as afforestation and transformation to agricultural land, both preceded by drainage, with long-term consequences for $N_2O$ emissions[2]. To reduce $N_2O$ emissions from wetland soils, we need a thorough understanding of biogeochemical pathways and critical environmental parameters, which shape the microbial activities underpinning the N cycle and $N_2O$ dynamics.

Microbial processes such as classical denitrification, nitrifier denitrification, and dissimilatory nitrate reduction to ammonia (DNRA) all contribute to $N_2O$ production mainly in anoxic conditions[3]. By contrast, ammonia oxidation, which is the first step in nitrification, is an aerobic process performed by three groups of ammonia oxidizing microorganisms: canonical ammonia oxidizing bacteria (AOB), ammonia oxidizing archaea (AOA), and complete ammonia oxidizers (comammox *Nitrospira*). AOA not only directly produce $N_2O$, but also provide substrate for denitrification[4]. Yet, little is known about the environmental conditions that favor each process and thereby $N_2O$ production and consumption. AOA may play a pivotal, underexplored role in fueling denitrification and facilitating terrestrial $N_2O$ emissions[5] in many soil environments.

Here we analyzed 645 wetland soils (Fig. 1a; Supplementary Data 1) to determine how the structure and function of microbial communities contribute to $N_2O$ emissions. Our unique dataset integrated global-scale analysis of functional metagenomes (to estimate relative abundance of N-cycle genes independently of PCR biases), multi-group metabarcoding (bacterial 16S, archaeal 16S, fungal 18S-ITS rRNA genes), absolute quantification of N-cycle gene abundances, as well as in situ $N_2O$ flux and ex situ potential $N_2$ production analyses. We further leveraged available genomics data to understand genetic mechanisms underlying $N_2O$ production. We hypothesized that the high $N_2O$ production in global wetland soils is mainly related to the diversity and abundance of nitrifying microbes, and that archaeal nitrifiers, both in terms of absolute and relative abundance to denitrifiers, are the most robust and accurate explanatory factor of $N_2O$ emissions from wetland soils globally.

## Results and discussion

**Global patterns of $N_2O$ fluxes**. Our analysis indicated that warmer soils and more intensive land use progressively may enhance $N_2O$ release from wetland soils. $N_2O$ emissions showed exponentially increasing relationships with temperature of the warmest month (Supplementary Fig. 1). In addition, the $N_2O$ emissions were strongly explained by land-use type ($r^2_{adj} = 0.364$, $p < 0.001$), with greatest values in the bare soils and lowest in the forest soils (Supplementary Fig. 2). Assessment of environmental determinants of $N_2O$ fluxes revealed that $N_2O$ emissions decline towards higher latitudes (Fig. 1, Supplementary Fig. 1). Contrary to the $N_2O$ emissions, potential $N_2$ production peaked in the temperate climate in negative correlation with land-use intensity (Supplementary Fig. 3). In agreement with our findings, a recent local warming experiment[6] and global models[2] predict an increase in $N_2O$ production in response to warming across various ecosystems.

**Relationships of global $N_2O$ fluxes to microbial diversity and taxa**. Our analyses of microbial communities of wetland soils revealed that, like the increasing $N_2O$ emissions towards the equator (Fig. 1b), archaeal diversity significantly increased towards low latitudes (Supplementary Fig. 4a). By contrast, mid-latitude wetland soils harbored the highest bacterial diversity, whereas fungal diversity showed no significant relationships with latitude but peaked at mean annual temperature of 10–15 °C (Supplementary Fig. 4). Across all associations among archaea, bacteria, and fungi of the wetland soils, climate and soil variables had the greatest impact on microbial diversity (Supplementary Fig. 5). General linear models combined with machine learning techniques indicated that archaeal diversity was best explained by soil C/N ratio, which agrees with a previous study on mineral soils[7]. Soil pH was the primary determinant of bacterial diversity (Supplementary Fig. 5) and relative abundance of the most common bacterial phyla (Fig. 2; Supplementary Fig. 6), whereas fungal diversity showed a weak relationship with environmental factors (Supplementary Fig. 5). These results corroborate those from mineral soils, where bacteria show stronger environmental associations than fungi and warm temperate regions harbor the highest bacterial diversity[8]. In addition, soil pH constitutes the main determinant of bacterial diversity in the mineral soil microbiome[8,9].

To determine the main microbial groups associated with $N_2O$ emission in global wetland soils, we related $N_2O$ fluxes with the relative abundance of various microbial lineages based on 16S and 18S rRNA gene metabarcoding. The microbial phyla Proteobacteria, Acidobacteriota, and Chloroflexi are the most abundant globally (Fig. 2). However, these groups were not significantly associated with $N_2O$ fluxes ($p > 0.05$), whereas the relative abundance of AOA from the phylum Thaumarchaeota emerged as the most strongly correlated group with $N_2O$ emission (Fig. 3). This is in agreement with a previous study on arctic peat soils, where the contribution of ammonia oxidizing archaea to $N_2O$ flux was confirmed by group-specific ammonia oxidation inhibitors as well as molecular approaches[10]. A previous study also reports a strong association between the thaumarchaeal 16S rRNA and *amoA* genes in environmental samples[11]. We also found that among all prokaryotic and eukaryotes genera uncovered in metagenomics data, the *Soil Crenarchaeotic Group* (SCG) showed the strongest positive correlation with $N_2O$ emissions (Supplementary Data 2). Furthermore, of the total 620 archaeal OTUs uncovered by a long-read sequencing technology (PacBio) occurring in >5 sites, 11 OTUs (including 5 in the order *Nitrososphaerales*, which are confirmed ammonia oxidizers; Supplementary Data 3) showed positive correlations ($r > 0.35$, $q < 0.2$) with $N_2O$ emission. Of these, $N_2O$ fluxes showed the strongest correlation with the relative abundance of OTUs most closely associated with 'Candidatus Nitrosotenuis chungbukensis MY2' ($r = 0.488$, $p < 0.001$) and 'Candidatus Nitrosocosmicus oleophilus MY3' (Spearman's rank-correlation $r = 0.477$, $p < 0.001$). Both taxa produce $N_2O$ in pure culture[12]. In agreement with our study, a previous study found that in arctic peatlands $N_2O$ emission was driven by only two OTUs of Thaumarchaeota, one of which was closely affiliated to 'Ca. N. oleophilus MY3'[10]. Ammonia oxidizing archaea play a key role in nitrification in unfertilized soils and soils with low ammonia concentrations[13]. In addition, in unfertilized soils, nitrite and nitrate may be predominantly made available for denitrifiers through nitrification, making nitrification a limiting factor for denitrification.

**Metagenomic analysis of pathways underlying global $N_2O$ fluxes**. To investigate functional pathways contributing to $N_2O$ emission, we examined clusters of orthologous gene groups (OGs) using metagenomes (see the "Methods" section). Among all potential key genes involved in $N_2O$ emission from archaea, the relative abundance of the archaeal *amoA* (ENOG411114F) showed

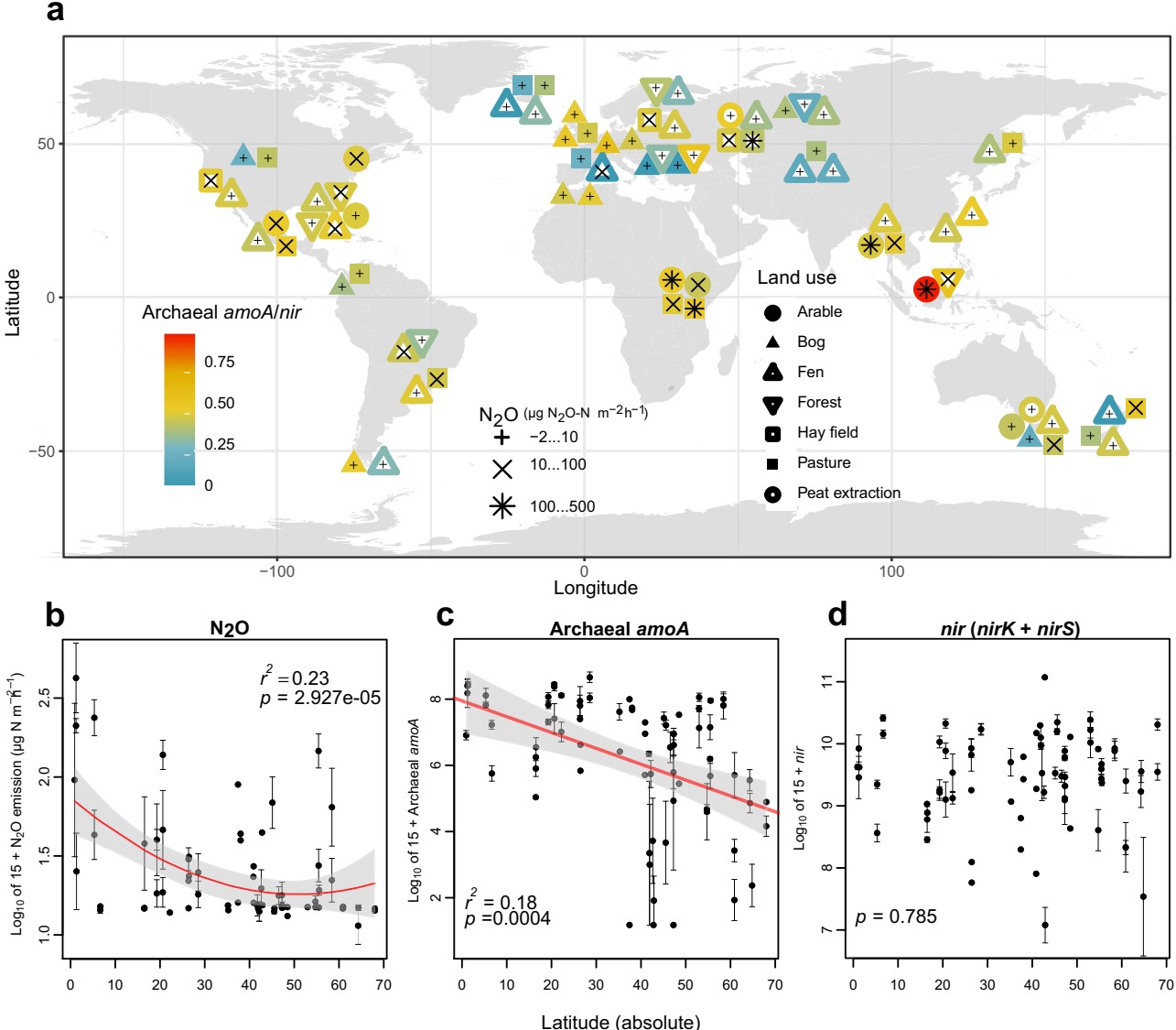

**Fig. 1 Global hotspots of N$_2$O fluxes in relation to archaeal nitrifiers across various land use types. a** Distribution of the study sites and their measured N$_2$O emissions as well as the archaeal-nitrifier/denitrifier ratio (archaeal *amoA*/(*nirK* + *nirS*)). Typographical symbols (+, ×, or *) denote average N$_2$O fluxes per site, the filled/open round, square, and triangle shapes represent different land-use types, and shape color shows the archaeal-nitrifier/denitrifier ratio based on the absolute abundance of gene copies determined by qPCR ($n = 72$ independent sites). **b–d** Latitudinal gradient of N$_2$O emissions, archaeal *amoA* and *nir* (*nirK* + *nirS*). Error bars represent the standard error (SE) of the means ($n = 74$ independent sites). The statistical test used was two-sided.

the strongest correlation with N$_2$O ($r = 0.625$, $p < 0.001$; Fig. 3), followed by an OG with unknown functions (Supplementary Data 4). To further evaluate the genetic basis facilitating N$_2$O emission, we compared the nucleotide sequences of the archaeal OTUs correlating with N$_2$O emission with those showing no such correlation. Using BlastN searches of 16S rRNA gene reads against complete archaeal genomes, we located the closest genome-sequenced relatives and obtained the corresponding genomic functional profiles. Based on these, we found that the aerobic ammonia oxidation pathway was restricted to four archaeal genera belonging to Thaumarchaeota—*Nitrososphaera*, *Nitrosocosmicus*, *Nitrosotenuis*, and *Nitrosarchaeum* (Supplementary Data 5). A strong association between the archaeal *amoA* gene abundance and N$_2$O emission occurred across both natural and disturbed sites. Soil nitrate (NO$_3^-$) content was also strongly correlated with the relative abundance of archaeal *amoA* ($r = 0.551$, $p < 0.001$). Comparative genomics analysis further revealed that archaea were more enriched in aerobic ammonia-oxidizing pathways compared

with bacteria (5.3% vs 0.3%; Supplementary Data 6–8). Overall, our results support the potential key role of Thaumarchaeota in N$_2$O emissions from wetland soils globally.

While the pathways and enzymes involved in thaumarchaeal N$_2$O production are not fully understood, it has been suggested that AOA can produce N$_2$O through both nitrosating hybrid formation and enzymatic denitrification[12,14]. Jung and colleagues proposed that '*Ca. N. oleophilus* MY3' has a denitrification capacity using the putative cytochrome P450 NO reductase, homologs of which are present in other representatives of the genus *Nitrosocosmicus*[12]. However, ammonia oxidizing archaea lacking these homologs are also able to produce N$_2$O[12,14]. Further studies are needed to establish the mechanisms behind thaumarchaeal N$_2$O production.

**Functional genes driving global N$_2$O fluxes.** To validate the observations from the metagenomic analysis and determine specific microbial genes involved in N$_2$O dynamics, we related

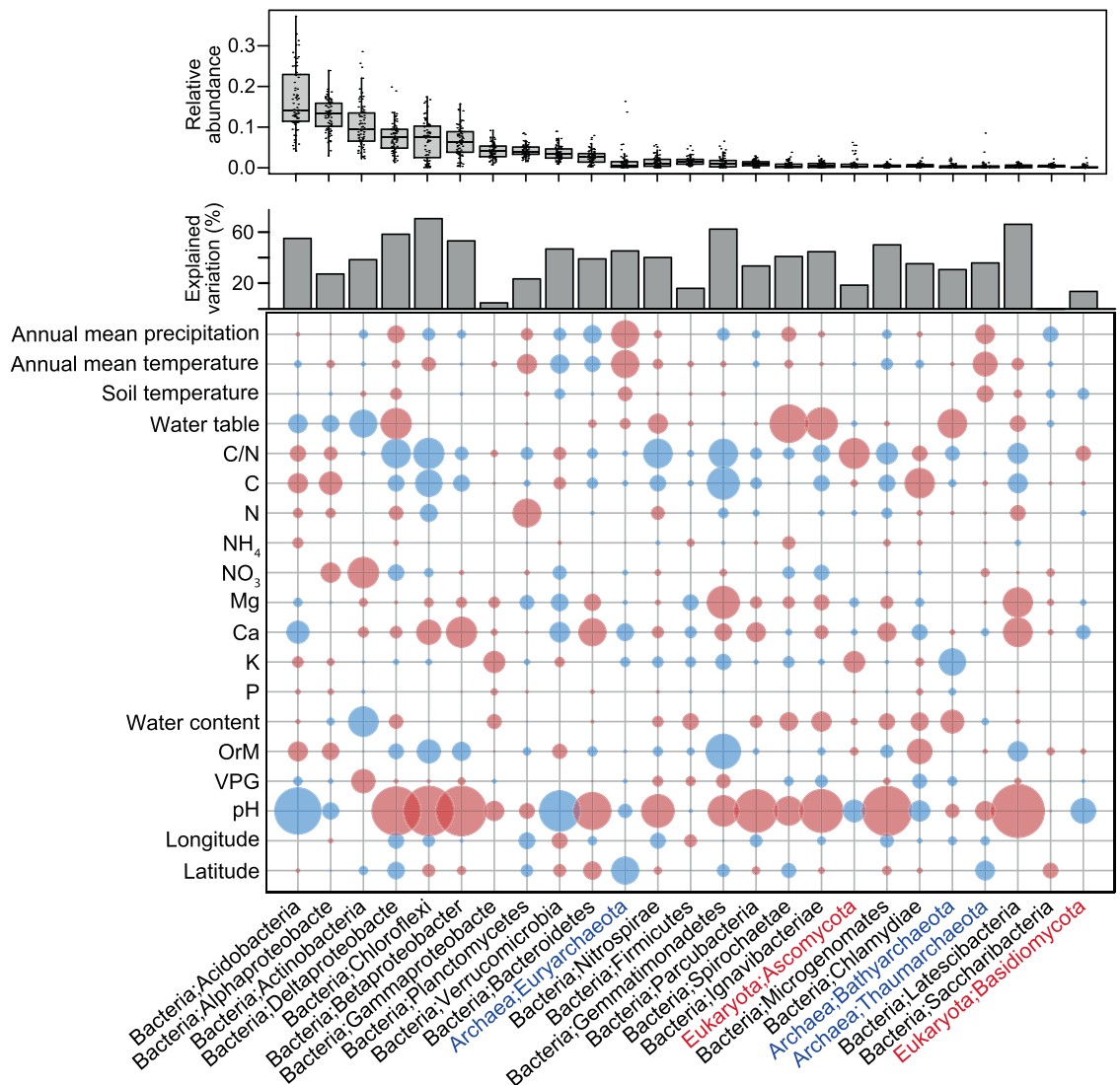

**Fig. 2 Environmental predictors of major archaeal, bacterial and fungal phyla (class for Proteobacteria) across the global wetland soils.** The relative abundance data are based on the relative abundance of SSU rRNA genes (normalized by total SSU rRNA abundances per sample) as revealed by shotgun metagenomics ($n = 74$ independent sites). Boxes represent 25th–75th percentile of the data distribution with whiskers at 1.5 × the interquartile range and the middle line representing median. The size of circles corresponds to the partial importance based on Random Forest models (variability% of mean decrease in accuracy estimated based on out-of-bag-CV); blue and *red* depict negative and positive Spearman correlations, respectively ($n = 74$ independent sites). Archaeal and fungal phyla names are indicated in blue and red colour, respectively. The abbreviations are organic matter (OrM), pH (soil pH), C/N (carbon to nitrogen ratio), Ca (calcium), K(potassium), P (phosphorous), Mg (magnesium), and Von Post grade of decomposition (VPG).

$N_2O$ emissions to the absolute abundance of main genes involved in the N cycle using quantitative polymerase chain reaction (qPCR). The abundance of archaeal ammonia monooxygenase (archaeal *amoA*; $r = 0.458$, $p < 0.001$) and bacterial *amoA* (bacterial *amoA*; $r = 0.313$, $p < 0.001$) had strongest positive correlations with $N_2O$ emission (Fig. 4, Supplementary Fig. 7). The relative increase in archaeal nitrifiers compared to denitrifiers in lower latitudes coincided with the greater $N_2O$ emissions in these regions (Fig. 1). The absolute archaeal *amoA* abundance was slightly higher than the bacterial *amoA* abundance (qPCR: $F = 6.00$, $p = 0.015$), substantiating the importance of archaea in nitrification across wetland soils (Supplementary Fig. 8), as previously reported for grassland and agricultural soils[15]. Our results also corroborate a local-scale metatranscriptomics study in mineral soils[16], suggesting that archaea predominate over bacteria for ammonia oxidation in soils.

Other major genes involved in the N cycle, including those known to be involved in $N_2O$ production, were surprisingly of

limited importance in explaining $N_2O$ emission (Fig. 3, Supplementary Fig. 7). The correlation between comammox *amoA* and $N_2O$ emission was expectedly weak, which may be related to the apparent adaptation of comammox *Nitrospira* to low ammonia or because comammox *Nitrospira* produce relatively small quantities of $N_2O$[17,18]. Furthermore, the absolute abundance of comammox *amoA* was lower than that of archaeal *amoA* (Supplementary Fig. 7). In addition, the abundance of reads related to anaerobic ammonium oxidation (anammox) and the nitrite/nitrate-dependent anaerobic methane oxidation (n-damo) did not correlate with the $N_2O$ fluxes. The abundance of *nosZ* genes, which encode the nitrous oxide reductase enzyme that consumes $N_2O$, was positively correlated with $N_2O$ emission (Supplementary Fig. 7). The denitrification genes responsible for $N_2O$ production (*nirK* and *nirS*) showed weak or no correlation with $N_2O$ emission (Supplementary Fig. 7). The abundance of *nir* genes was strongly correlated with that of *nosZ* genes (Fig. 4a) that reduce $N_2O$ into inert $N_2$. This consumption may explain

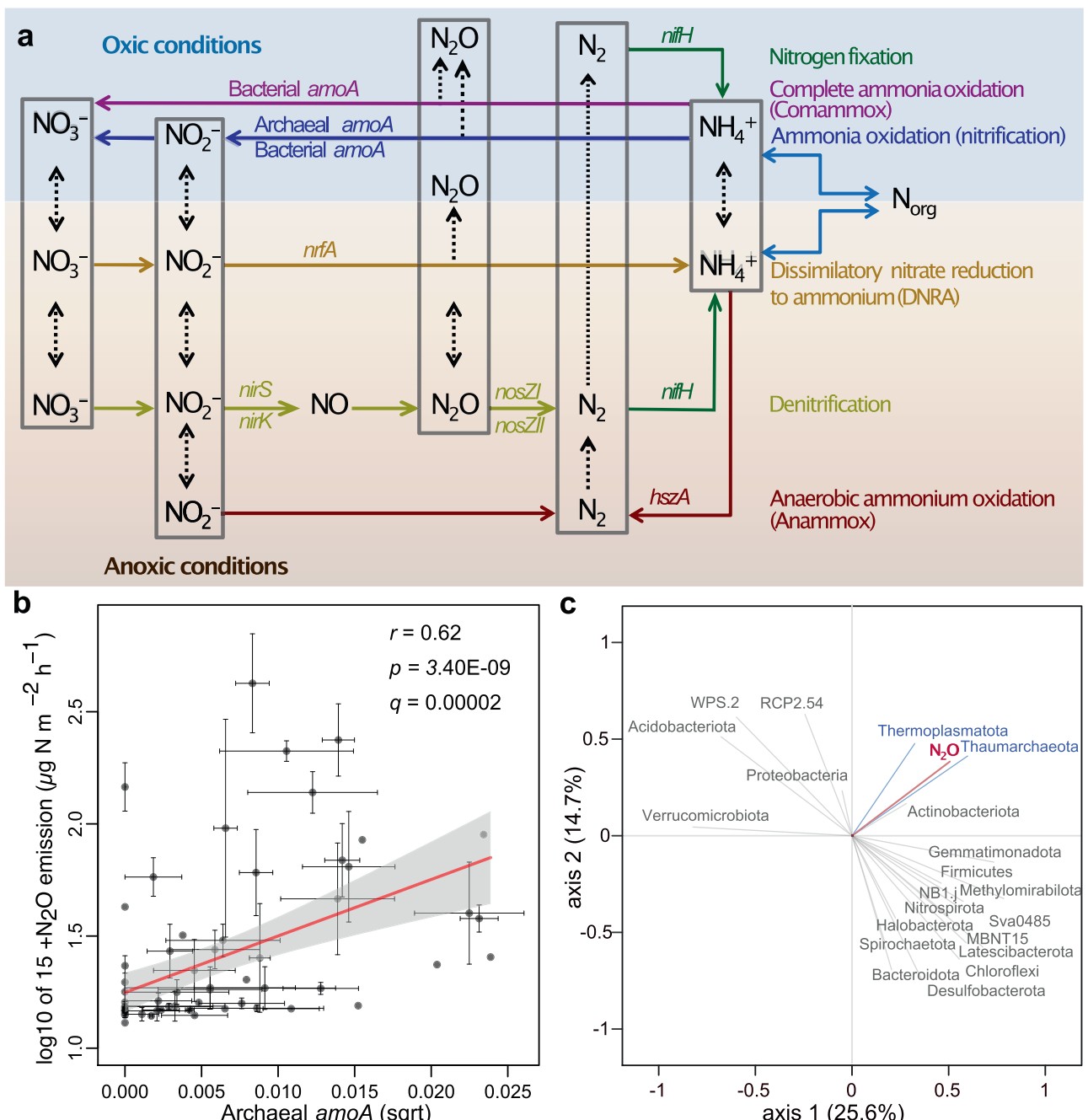

**Fig. 3 Archaea and archaeal *amoA* strongly correlate with N₂O across global wetland soils. a** Schematic view of nitrogen cycle in soils and the key genes involved. **b** Relationship between site mean relative abundance of archaeal *amoA* and N₂O emission ($n = 74$ independent sites). The relative abundance of archaeal *amoA* was determined based on the relative abundance of metagenomics reads assigned to ENOG411114F (extracted from Hellinger transformed abundance matrix of archaeal OGs). The inset numbers represent a Spearman rank correlation coefficient (*r*) and corrected *p*-value for multiple testing using Benjamini–Hochberg method (*q*). Error bars represent the standard errors (SE) of the site means. **c** Partial least-squares regression (PLS regression) plot showing the relationships among the relative abundances of prokaryotic taxonomic groups (as determined by 16S metabarcoding) and N₂O emission ($n = 74$ independent sites). Blue lines represent archaeal phyla. The statistical test used was two-sided.

low N₂O emissions from the soils enriched with *nir* genes. Potential N₂ production, however, was not significantly correlated with *nosZ* abundance (Fig. 4a). In addition, the set of genes associated with denitrification may vary in different species, and not all denitrifiers possess all genes related to this process[19,20]. Soil pH and organic carbon concentration may also affect the amount of N₂O produced from denitrification (Supplementary Fig. 5). Nevertheless, denitrifiers may be more metabolically versatile than nitrifiers and use a range of compounds for both

energy and respiration, which is reflected in their weaker environmental associations (Supplementary Figs. 2, 5, 9). There have been many previous attempts to correlate denitrification genes with soil N₂O fluxes; whilst some studies have found a good correlation[21], others have not[22–25].

Next, we related N₂O emissions with the diversity of all major genes involved in the N cycle (based on their absolute abundances quantified by qPCR) and found higher N₂O emissions with increasing diversity of N cycle functional genes (Fig. 4c). The

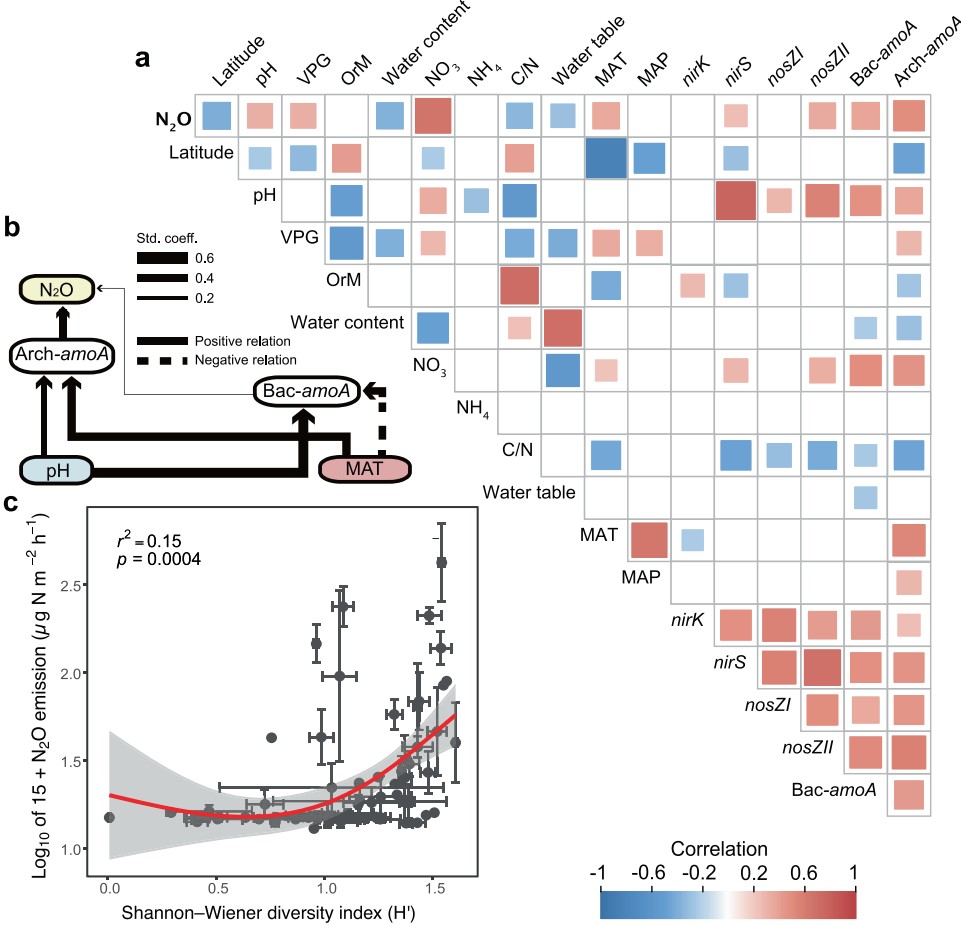

**Fig. 4 Nitrogen-cycle genes as the main factors explaining N₂O emissions across the global wetland soils. a** Correlations between environmental variables, the abundance of *nir*, *nosZ* and *amoA* genes (quantified by qPCR) and N₂O emission ($n = 74$). The abbreviations are archaeal *amoA* (arch-*amoA*), bacterial *amoA* (bac-*amoA*), organic matter (OrM), pH (soil pH), C/N (soil carbon to nitrogen ratio), Von Post grade of decomposition (VPG). **b** Structural equation modeling (SEM) showing niche differentiation between bacterial and archaeal *amoA* ($n = 74$ independent sites). The model fitness was acceptable (Fisher's $C = 8.4$, $p = 0.08$). Line thickness corresponds to standardized regression coefficients as indicated in the legend. Dash lines indicate negative relationships. The statistical test used was two-sided. The abbreviations are mean annual temperature (MAT), pH (soil pH). **c** Relationship between N₂O emissions and the diversity of N cycle functional genes that are directly involved in N₂O dynamics, including archaeal *amoA*, bacterial *amoA*, comammox *amoA*, *nirK*, *nirS*, *nrfA*, *nosZI*, and *nosZII*. The inset numbers represent an adjusted $r^2$ and $p$-value from a GAM model. Error bars represent the standard error (SE) of the means ($n = 74$ independent sites). The statistical test used was two-sided.

greater N₂O emissions with increased functional gene diversity can be related to the functional complementarity of different N-related processes, in particular denitrification and nitrification, in producing N₂O. This can occur in drained wetland soils with a variety of anoxic and oxic conditions[26], where nitrifiers may contribute to the generation of nitrate required for denitrification[27]. Variability of environmental conditions governed by water level dynamics has been shown to determine the diversity of microbes that affect N₂O fluxes in wetland soils[28,29]. However, we found that the effect of soil factors (C/N ratio and pH) and temperature on functional gene diversity in our dataset (collectively explaining 57% of the variation; Supplementary Figs. 5, 9, 10) exceeded that of soil water content. Among the studied functional genes, the abundance of archaeal *amoA* correlated the best to temperature and C/N ratio (Supplementary Fig. 5), similarly to the N-cycle gene diversity. Contrary to our expectation, taxonomic diversity of microbes showed no correlation with N₂O emissions ($p > 0.05$). Previous studies have shown conflicting results on the relationship between microbial diversity and N₂O emissions, as reviewed in ref. [30]. The decoupling of taxonomic and functional diversities may be due to functional redundancy in microbial taxa active in the N cycle[31].

**Environmental determinants of N₂O related microbial communities.** We explored the environmental conditions favoring microbial taxa and genes driving N₂O emission. The archaeal *amoA* displayed a unimodal relationship with mean annual air temperature peaking around 20 °C ($r^2_{adj} = 0.255$, $p < 0.001$; Supplementary Fig. 9). This supports earlier findings of greater AOA activities in warmer seasons[32]. The strong positive correlations of the AOA/AOB ratio, mean annual air temperature and soil temperature (Supplementary Fig. 11) is in line with the relatively high temperature optimum of AOA[33]. This finding suggests that elevated (>15 °C) soil temperature in combination with optimal soil moisture[31] may promote N₂O emissions from soils due to an increased AOA abundance. Nevertheless, how this may be offset by their altered balance with AOB remains to be determined.

**Implications for predicting global N₂O fluxes.** Our analyses indicate that both the structure and function of wetland soil microbiome and climatic conditions determine N₂O fluxes globally. Considering the combined effect of optimal soil moisture and temperature, archaea are important contributors to N cycling in drained wetland soils[34]. Furthermore, we provide evidence that

archaeal abundance is a key factor associated with ammonia oxidation pathway that underlies $N_2O$ emission in wetland soils globally. Our results complement previous findings on the major role of archaea in $N_2O$ emission in alpine soils[35] and oceans[36]. In particular, the global distribution of AOA and their adaptation to low oxygen and ammonia concentrations may be suggestive of the substantial role of this microbial group in N cycling of wetland soils.

Taken together, our results suggest that nitrifying microbes may contribute more strongly to $N_2O$ emission than previously thought, and that the diversity of microbes involved in the N cycle may be the integral predictor of $N_2O$ emissions. To determine the mechanisms underlying global $N_2O$ emissions, we need to understand the relative role of nitrification and denitrification across a broad variety of habitat types as well as the effects of climate, vegetation, and land use. We predict that future drainage and warming of wetland soils will have negative consequences for regulating ecosystem services of wetlands through accelerating archaeal nitrification that increases substrate availability for denitrification, which collectively promote $N_2O$ emission. Although we could not distinguish cause and effect, our study generates insights into nitrogen cycling and microbial drivers of $N_2O$ emission in wetlands.

## Methods

**Study sites and sampling.** We sampled gas and soil in 29 regions throughout the A (rainy tropical), C (temperate), and D (boreal) climate types of the Köppen classification from six continents during the vegetation period between August 2011 and June 2018, following a standard protocol[26]. According to the protocol, the gas and soil samples were collected from locations in public domain or in previous agreement with the local community and/or property owner. The samples were exported from the origin countries and imported to Estonia, EU in cooperation with customs officers of the respective states, following the legal provisions of soil export and import, specifically exemptions for scientific purposes. To capture the full range of environmental conditions in each region, we established 76 wetland soil sites under different vegetation (mosses, sedges, grasses, herbs, trees, and bare soil) and land-use types (natural—raised bog, fen, and forest; agricultural —arable, hay field and pasture; and a peat extraction area) (Fig. 1a; Supplementary Data 1). We used a four-grade land-use intensity index to quantify the effect of land conversion: 0, no agricultural land use (natural mire, swamp, or bog forest); 1, moderate grazing or mowing (once a year or less); 2, intensive grazing or mowing (more than once a year); and 3, arable land (directly fertilized or unfertilized). The vegetation and land-use intensity types and the land-use intensity index were estimated from observations and contacts with site managers and local researchers.

Within the sites, we established 1–4 stations 15–500 m apart to maximize the captured environmental variation. Each of the 196 stations was equipped with 3–5 opaque PVC 65 L truncated conical chambers 1.5–5 m apart and an observation well (perforated, 50 mm diameter PP-HT pipe wrapped in geotextile; 1 m in length). From each of the 645 chambers, $N_2O$ fluxes were measured following the static chamber method[37] using PVC collars (0.5 m diameter, installed to 0.1 m depth in soil). Stabilization of 3–12 h was allowed before gas sampling to reduce the disturbance effect of inserting the collars on fluxes. The chambers were placed into water-filled rings on top of the collars. Gases were sampled from the chamber headspace into a 50 mL glass vial every 20 min during a 1-h session. The vials had been evacuated in the laboratory 2–6 days before the sampling. At least three sampling sessions per location were run within 3 days. Water-table height was recorded from the observation wells during the gas sampling at least 8 h after placement. Soil temperature was measured at the 10 and 20 cm depth.

Soil samples of 150–200 g were collected from the chambers at 0–10 cm depth after the final gas sampling, and transported to laboratories in Tartu, Estonia. The homogenized samples were divided into subsamples for physical–chemical analyses and DNA extraction. The samples for chemical analyses were stored at 4 °C and microbiological samples were stored at –20 °C. DNA extraction was provided at the Tartu University environmental microbiology laboratory (see details below). Using a PP-HT plastic cylinder, intact soil cores (diameter 6.8 cm, height 6 cm) for the $N_2$ analysis with the $He$–$O_2$ method[38] were collected from the topsoil (0−10 cm) inside 252 chambers at 26 sites, starting from 2014. Samples from different climates were run at respective temperatures. During transport, the soil samples were kept below the ambient soil temperature from which they were collected.

**Gas flux analyses.** The gas samples were analyzed for $N_2O$ concentration within 2 weeks using two Shimadzu GC-2014 gas chromatographs equipped with ECD, TCD, and a Loftield-type autosampler. The $N_2O$ fluxes were determined on linear regressions obtained from consecutive $N_2O$ concentrations taken when the chamber was closed, using $p < 0.05$ for the goodness of fit as a quality threshold for

the linear regression. During the quality control, in cases of insignificant regression ($p > 0.05$ we removed one outlier. If the regression remained insignificant but the flux value fell below the gas-chromatography measuring accuracy (regression change of $N_2O$ concentration $\delta v < 10$ ppb), we included it in the subsequent analyses as a zero value.

The helium atmosphere soil incubation technique[30] was used to measure potential $N_2$ fluxes from soil cores. The cylinders with intact soil cores were placed into special gas-tight incubation vessels located in a climate chamber. Gases were removed by flushing with an artificial gas mixture (21.0% $O_2$, 358 ppm $CO_2$, 0.313 ppm $N_2O$, 1.67 ppm $CH_4$, 5.97 ppm $N_2$, and He). The new atmosphere equilibrium was established after 12–24 h by continuously flushing the vessel headspace with the artificial gas mixture at 20 mL/min. The flushing time depended on the soil moisture. Incubation temperature was kept similar with the field conditions. The gas-chromatograph (Shimadzu GC-2014) equipped with a thermal conductivity detector was used to measure $N_2$ concentration in the mixture of emitted gases accumulated in the headspace (start value, 40, 80, and 120 min as final value) of the cylinder after 2 h of closure. The gas concentration in the chambers increased in a near-linear fashion and linear regression was applied for calculation of the fluxes. The flux measurements with $r^2$ of 0.81 ($p < 0.1$) or greater were used.

**Soil physico-chemical analysis.** Plant-available phosphorus (P, $NH_4$-lactate extractable) was determined on a FiaStar5000 flow-injection analyzer. Plant-available potassium (K) was determined from the same solution by the flame-photometric method and plant-available magnesium (Mg) was determined from a 100 mL $NH_4$-acetate solution with a titanium-yellow reagent on the flow-injection analyzer. Plant-available calcium (Ca) was analyzed using the same solution by a flame-photometric method. Soil pH was determined using a 1 N KCl solution; soil $NH_4−N$ and $NO_3−N$ were determined on a 2 M KCl extract of soil by flow-injection analysis (APHA-AWWA-WEF, 2005). Total nitrogen and carbon contents of oven-dry samples were determined by a dry-combustion method on a varioMAX CNS elemental analyzer (Elementar Analysensysteme GmbH, Germany). Organic matter content of dry matter was determined by loss on ignition. We determined soil water content (SWC) from dry matter content and empirically established bulk densities of mineral and organic matter fractions.

**DNA extraction, DNA library preparation, and sequencing.** DNA extraction was performed from 0.2 g of frozen soil samples (homogenized) using the Qiagen DNeasy PowerSoil Kit (12888-100), following the manufacturer's recommendations. DNA concentrations were measured with Qubit™ 1X dsDNA HS Assay Kit using Qubit 3 fluorometer (Invitrogen). Altogether 645 individual soil samples were selected for metabarcoding of bacteria, archaea, and eukaryotes. For bacteria, we used the primers 515F (5′-GTGYCAGCMGCCGCGGTAA-3′) and 806RB (5′-GGACTACNVGGGTWTCTAAT-3′) to amplify the variable V4 region of the 16S rRNA gene[39]. Although these primers co-amplify archaea to some extent, we sought to specifically amplify a longer portion of their 16S rRNA gene to capture their full diversity, using the primers SSU1ArF (5′-TCCGGTTGATCCYGCBRG-3′) and SSU1000ArR (5′-GGCCATGCAMYWCCTCTC-3′)[40]. To amplify a broad range of eukaryotes, we used the primers ITS9mun (5′-GTACACACCGCCCG TCG-3′) and ITS4ngsUni (5′-CGCCTSCSCTTANTDATATGC-3′) that cover the V9 variable region of the 18S rRNA gene and the full internal transcribed spacer (ITS) region[41]. Both the forward and reverse primers were tagged with a 12-base multiplex identifier (MID), except in the case of archaea where only the forward primer was tagged with MID. All PCRs were performed in two replicates using 5 × HOT FIREPol® Blend Master Mix (Solis BioDyne, Tartu, Estonia) in 25 µl volume. By default, the bacteria, archaea, and eukaryotes were amplified using 25, 35, and 30 cycles, respectively. In case of no amplification, two or five extra cycles were added, or DNA was re-extracted and re-purified. Thermal cycling included an initial denaturation at 95 °C for 15 min; 25–40 cycles of denaturation for 30 s at 95 °C, annealing for 30 s at 55 °C, elongation for 1 min at 72 °C; final elongation at 72 °C for 10 min; and storage at 4 °C. The two replicates of each reaction were pooled and visualized on TBE 1% agarose gel.

The bacterial amplicons were sequenced using the Illumina NovaSeq platform at 2 × 250 bp paired-end mode. Illumina amplicon libraries were generated using TruSeq DNA PCR-Free High Throughput Library Prep Kit with TruSeq DNA CD Indexes (Illumina). To increase identification accuracy and coverage, the archaeal and eukaryote amplicons were sequenced using a long-read sequencing technology on PacBio Sequel II platform[40,41]. SMRTbell library preparation followed the Pacific Biosciences Amplicon library preparation protocol. Metabarcoding analysis was repeated for samples yielding <5000 prokaryotic reads (Illumina), <500 archaeal reads (PacBio), or <1000 eukaryote reads (PacBio).

For the functional metagenome analysis, three replicate soil samples per station were pooled based on equimolar amount of DNA. Library preparation and indexing of each 196 pooled samples was performed using Nextera XT DNA Library Prep Kit in combination with Nextera XT Index kits v2 (Illumina). Metagenomes were sequenced based on the shotgun approach to an expected depth of 5,000,000 reads using Illumina NovaSeq with 2 × 150 bp paired-end mode. The samples with <1,000,000 reads were subjected to resequencing.

**Quantitative PCR**. We used qPCR to quantity the absolute abundance of bacterial and archaeal 16S rRNA genes as well as the key genes involved in N cycle pathways, including denitrification (*nirS*, *nirK*, *nosZ* clade I, and *nosZ* clade II), N fixation (*nifH*), dissimilatory nitrate reduction to ammonia (DNRA; *nrfA*), ammonia oxidation (bacterial *amoA*, archaeal *amoA*, comammox *amoA*), and anammox- and n-damo-specific 16S rRNA genes (Supplementary Fig. 7). The qPCR assays were performed using RotorGene® Q equipment (Qiagen, Valencia, CA, USA). The qPCR method was performed following ref. [34]. Briefly, the qPCR reactions were performed in 10 μL volume containing 5 μL Maxima SYBR Green Master Mix (Thermo Fisher Scientific Inc., Waltham, MA, USA), an optimized concentration of forward and reverse primers, 1 μL of template DNA and sterile distilled water. The gene-specific primer sets, optimized primer concentrations and thermal cycling conditions for each target gene are shown in Supplementary Data 8. The quantification data were analyzed with RotorGene Series Software (version 2.0.2; Qiagen, Hilden, Germany) and LinRegPCR program (version 2020.0) [42]. The gene abundances were calculated as a mean of fold differences between a sample and each 10-fold standard dilution in respective standard as recommended by ref. [42]; gene abundances were reported as gene copy numbers per gram of dry soil.

### Bioinformatics

*Metabarcoding.* Illumina MiSeq sequences were analyzed using LotuS software[43] following ref. [44]. Briefly, the reads were demultiplexed and quality-filtered by trimming individual reads to 170 bp and removing reads with an accumulated error >2 or an estimated accumulated error >2.5 at a probability of ≥0.01. To pass to the next step, each unique read (reads preclustered at 100% identity) was required to be present at least eight times in at least one sample, four or more times in at least two samples, or three or more times in at least three samples. Chimeric OTUs were removed based on both reference-based and de novo chimera checking algorithms as implemented in uchime[45]. The resulting OTUs were taxonomically annotated by aligning their sequences with Lambda[46] to SILVA v135 database[47] and the LotuS least common ancestor (LCA) algorithm (options: -p miSeq derepMin 8:1,4:2,3:3 –simBasedTaxo 2 –refDB SLV -thr 8). For processing PacBio sequencing data, PipeCraft[48] was used as follows. Raw sequencing data was demultiplexed via *mothur* (version 1.36.1)[49] module in PipeCraft by allowing one mismatch to tag region (i.e. to index sequence that was used for multiplexing); quality filtering was performed using vsearch (version 1.11.1)[50] module with maximum expected error threshold of 1 (--fastq_maxee = 1) and discarding sequences with ambiguous bases (--fastq_maxns = 0); putative chimeric reads were discarded using vsearch uchime_denovo algorithm; prior clustering, full length ITS reads without conservative regions (18S and 28S rRNA genes; i.e. primer-binding sites) were extracted using ITSx software (version 1.0.11)[51]; using UPARSE (version 8.1.1861), sequences were clustered to OTUs at 98% sequence similarity where singletons (clusters with only one sequence) were removed during the process (minsize = 2). Representative sequences of OTUs were taxonomically annotated based on the best blast hit against UNITE database (version 8)[52] followed by the LCA algorithm. For statistical analyses, we retained 645, 440, and 638 samples that yielded sufficient sequencing depth for Illumina 16S data (bacteria and archaea), PacBio 16S data (archaea) and PacBio ITS (fungi), respectively.

*Metagenomics.* Analysis of metagenomic reads was done using MATAFILER pipeline[53]. Briefly, reads obtained from the shotgun metagenomic sequencing of peat samples were quality-filtered by removing reads shorter than 70% of the maximum expected read length (150 bp), with an observed accumulated error >2 or an estimated accumulated error >2.5 with a probability of ≥0.01, or >1 ambiguous position. Using sdm software (version 1.46)[43], reads were trimmed if base quality dropped below 20 in a window of 15 bases at the 3′ end, or if the accumulated error exceeded 2. Altogether 196 samples produced sufficient quantity of reads and were retained for statistical analyses. To estimate the functional composition of each sample, we implemented a similarity search approach using DIAMOND (version 2.0.5; options -k 5 -e 1e-4 –sensitive) in blastx mode[54]. Prior to that, the quality-filtered read pairs were merged using FLASH (version 1.2.10)[55]. The mapping scores of two unmerged query reads that mapped to the same target were combined to avoid double counting. In these cases, the hit scores were combined by averaging the percent identity of both hits. The best hit for a given query was based on the highest bit score and highest percent identity to the subject sequence. Using this method, we calculated the relative abundance of (clusters of) orthologous gene groups (OG) by mapping quality-filtered reads against the eggnog database (version 4)[56]. We also calculated metagenomic relative abundances (i.e. miTag[57]) of different taxonomic groups based on small subunit (SSU) rRNA genes. For this, SortMeRNA (version 2.0)[58] was used to extract and blast search rRNA genes against the SILVA SSU database (v128). Reads approximately matching this database with $e < 10^{-4}$ were further filtered with custom Perl and C++ scripts, and merged using FLASH. In case read pairs could not be merged, the reads were interleaved such that the second read pair was reverse complemented and then sequentially added to the first read. Of these preselected reads, 50,000 reads were fine-matched the Silva SSU database using Lambda and the lowest common ancestor (LCA) algorithm adapted from LotuS.

*Genomic analysis.* The taxonomic analysis revealed that a few microbial lineages may disproportionally outperform the community functional diversity of microbes in affecting ecosystem biogeochemistry. Thus, we followed a trait-based approach to confirm our findings. We downloaded 385 complete archaeal genomes from NCBI as of 10/7/2020 (search terms: archaea[Organism] AND "complete genome"). These were used to build a reference database for a BlastN search to identify corresponding genomes and functional annotations of our archaeal OTUs. In addition, to better understand the potential functions of different archaeal lineages in N cycling, the functional annotation of all available archaeal genomes was retrieved from the Integrated Microbial Genomes and Microbiomes database (img.jgi.doe.gov) as of 15/7/2020.

**Data analysis**. To account for differences in sequencing depth across samples, diversity indices (Shannon diversity index) were calculated based on rarefied abundance matrices (metabarcoding datasets) in *vegan* package[59] of R (version 2.5-6). Multivariate analyses were performed using Bray-Curtis dissimilarity on normalized taxa abundance matrices in *vegan*. All raw P-values of multiple tests were corrected using Benjamini–Hochberg method. Taxonomic abundance data were normalized using Hellinger transformation as implemented in *vegan*.

To test the effect of biotic variables on N₂O emissions, we used Spearman correlation analysis components to identify the bacterial and archaeal taxonomic lineages and fungal OTUs most strongly associated with N₂O emissions. Functional gene (OG) composition and taxonomic community matrices were normalized by library size using Hellinger transformation. We subsequently used partial least squares (PLS) analysis to predict N₂O emissions based on taxonomic groups, which allows the dimensionality of multivariate data to be reduced into PLS components using *plsdepot* package[60] of R (version 0.1.17). Prior to this, we performed a backward variable elimination procedure to remove variables with low explanatory power (VIP threshold < 1), as implemented in *plsVarSel* package[61] of R (version 0.9.6).

For univariate analysis, the best predictors of the diversity and relative abundances of taxonomic and functional groups were identified using a machine learning approach implemented in *randomForest* package[62] of R (version 4.6-14). To further test direct and indirect effects of variables in the best model, structural equation modeling (SEM) was used as implemented in *piecewiseSEM* package[63] of R (version 2.1.0). The prior model was constructed based on our hypothesis (see the section "Introduction"). The optimal model fit was achieved by subsequent iterative revision based on modification indices. For linear relationships, Spearman's rank-correlation coefficient was calculated in R. For fitting non-linear relationships between soil water content and N₂O emissions, generalized additive model (GAM) were constructed using smoothing parameter estimated by marginal likelihood (REML) maximization, as implemented in mgcv package[64] of R (version 1.8-33). We also compared the goodness of fit estimates between first and second order polynomial models for certain analyses. The best polynomial fit was determined on the basis of Akaike Information Criterion (AIC) scores using "AIC" and "poly" functions of R.

## Data availability

All metabarcoding sequences and associated metadata have been deposited in the European Bioinformatics Institute Sequence Read Archive database: https://www.ncbi.nlm.nih.gov/bioproject/PRJNA718418; metagenomics sequences and associated metadata have been deposited at The European Nucleotide Archive under accession number https://www.ebi.ac.uk/ena/browser/view/PRJEB44414. Additional data generated in this study are provided in the Supplementary Information/Source Data file. SILVA database is available at https://www.arb-silva.de; UNITE database is available at https://unite.ut.ee/repository.php; Integrated Microbial Genomes is available at https://img.jgi.doe.gov; eggnog database is available at http://eggnog5.embl.de/download/eggnog_4.0/ Source data are provided with this paper.

## Code availability

The pipeline to process metabarcoding samples is available under https://psbweb05.psb.ugent.be/lotus/downloads.html and https://doi.org/10.15156/bio/587450. The pipeline to process shotgun metagenomic samples is available under https://github.com/hildebra/MATAFILER (https://doi.org/10.5281/zenodo.5831723).

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

## Acknowledgements
We thank O. Botsarova, L. Lopp, K. Kanger, R. Puusepp, E.J. Sova, H. Tamm, I. Varik for assistance in molecular laboratory analyses. We thank Lorenzo Menichetti for useful discussions. We are grateful to S. Egorov, I. Filippov, G. Gabiri, J. Gallagher, I. Gheorghe, W. Hartman, R. Iturraspe, J. Järveoja, A. Kull, F. Laggoun-Défarge, E. Lapshina, A. Lohila, C. Luswata, S. Mander, M. Metspalu, W. Mitsch, R. Moreton, K. Oopkaup, H. Óskarsson, J. Paal, T. Pae, E. Parrodi, S. Pellerin, F. Sabater, J. Salm; F. Sgouridis, D. Silveira Batista, K. Sohar, K. Storey, M. Tenywa; S. Ullah, E. Uuemaa, G. Veber, J. Villa, L. Yang and S.S. Zaw for assistance on study-site selection and field investigation. ME, JP, LT and UM were supported by the Ministry of Education and Science of Estonia (SF0180127s08 grant), the Estonian Research Council (IUT2-16, PUTJD619, PRG-352, PRG-609, and MOBERC20), EU through the European Regional Development Fund (Centres of Excellence ENVIRON, grant number TK-107, EcolChange, grant number TK-131, and the MOBTP101 returning researcher grant by the Mobilitas Pluss program); M.B. was supported by the Swedish Research Council Formas (2020-00807); J.P., K.K., and M.Ma. were supported by the European Social Fund (Doctoral School of Earth Sciences and Ecology); L.L.M. was supported by Royal Society Dorothy Hodgkin Research Fellowship (DH150187); F.H. was supported by European Research Council (ERC) Starting Grant (UNITY 852993), the BBSRC Institute Strategic Program Gut Microbes and Health BB/r012490/1, its constituent project BBS/e/F/000Pr10355, the European Union's Horizon 2020 research and innovation program (grant agreement no. 948219).

## Author contributions
Ü.M., J.P., and M.E. set up and conducted the field experiments. L.T. supervised DNA extraction and sequencing analysis. F.H. processed metagenomics data. S.A. processed metabarcoding data; M.E. performed qPCR analysis. M.B., M.E. and J.P. analyzed the data. L.L.-M. revised taxonomic and functional annotations. M.B. wrote the first draft of the manuscript with input from M.E., J.P., L.T. and Ü.M. The manuscript was revised by M.B., M.E., J.P., L.L.-M., K.K., U.K., J.L., M.Ma., M.Mo., U.N., M.O., M.P., K.S., M.Z., F.H., L.T., and Ü.M. All authors approved the submitted version.

## Funding

## Competing interests
The authors declare no competing interests.
