## [Peer Review File · Nature Communications]

REVIEWER COMMENTS

Reviewer #1 (Remarks to the Author):

This study examines wetland microbial community structure along latitudinal gradients, and relates the abundances of specific taxa and functional genes to nitrous oxide fluxes. Primarily by using correlation analyses, the study posits that archaea play a more important role in wetland N₂O emission than previously recognized. The greatest strength of this study is the sample size as well as the vast geographic coverage enabling comparisons across wetland systems under different habitat and land use conditions. The paper is generally well-written. Methods are generally robust and are explained with sufficient detail. Discussion of the results can often be improved by providing better context (please see specific comments below).

My main concern with this study is that I do not believe the conclusions as stated are sufficiently supported by the data presented. For example, the two primary takeaways are that: (i) archaea drive N₂O production in wetlands; and (ii) the greater the diversity of N-cycling functional groups, the greater the N₂O fluxes. Since this study design only involved correlation-based analyses, I think the authors should refrain from conclusively assigning causation. I apologize if I missed something, but it appears to me that the above conclusions are derived solely from the fact that N₂O values correlated the best with (i) archaeal abundances and (ii) the Shannon diversity index for N-cycling functional genes. Mechanistic explanations/hypotheses, if any, are not given for either of the two points. The association between AOA and N₂O fluxes is really interesting, and has not been identified prior to this study. However, I don't think there is sufficient evidence to state that AOA drive N₂O fluxes just based on the correlations.

In general, I would like to see better contextualization of the results, acknowledging the limitations of the study design.

Specific comments:

Line 31: Contradicts with the first sentence where wetlands are highlighted as the greatest source of N₂O.

Line 33-34: A minor but important distinction – the authors examined the functional potential of microbial communities, not the 'gene functions'

Line 34: Please include the actual number instead of '>600'

Line 37: I'm not sure what the authors mean by the 'diversity of N-cycle processes driven by nitrifying microbes.' Please note that the term 'nitrifiers' include both ammonia and nitrite oxidizers (I assume the authors are specifically referring to ammonia oxidizers here). Still, these organisms do not mediate a diverse range of N-cycle processes – please be more specific as to which processes the authors are referring to.

Line 39: While I agree N₂O fluxes correlated best with archaeal abundances, I'm not convinced that this

is 'unambiguous evidence' for a dominant role of archaea in N₂O production. This study has not tested causal links between wetland archaea and N₂O production.

Line 52 and throughout: Please check the order of the references. For example, in line 52, the reference should be numbered 2, not 5.

Line 60-61: Please add references.

Line 68: I would suggest changing 'N cycling' to 'N₂O emissions'

Line 77: Which figure shows the results of testing archaeal diversity as a predictor of N₂O fluxes? Does this refer to AOA diversity alone, or the archaeal phylum? If there is a correlation here, how could it be explained mechanistically?

Lines 94-99: Please discuss these findings in the context of previous studies.

Line 98: The water table depth? Also, looks like this is only true for SAP, and even then, the variation explained is almost close to zero?

Lines 103-104 and throughout: Please always use the complete gene names. This should be 16S and 18S rRNA gene metabarcoding.

Line 105: I'm not sure if the Extended figure 5 shows this. Is there another figure showing the phyla abundances?

Line 107-108: Based on Fig. 2, how are the authors concluding that AOA/Thaumarchaeota are the main predictor of N₂O fluxes? They co-vary, as can be seen in the PLS regression plot, but isn't that correlation only? Did the authors perform a separate prediction task that is not plotted in Fig. 2?

Line 114: Change 'mostly costly' to 'most costly'

Lines 114-116: Do the r and p values in parentheses refer to OTU correlations with N₂O fluxes? The placement is a bit confusing.

Lines 126-127: How do the results stated in this paragraph show that nitrification activity is associated with AOA relative abundance since activity was not measured in this study?

Lines 132-133: Figure 1 does not provide evidence for archaeal dominance in the tropics. It would be really helpful to have the raw qPCR abundances for these genes since ratios can obscure meaning. For instance, here the AamoA/(nirK+nirS) ratio could be higher in the tropics either due to an increase in AOA numbers or due to a lower abundances of nirK or nirS-harboring denitrifiers.

Lines 132-133 and Figure 1: Why did the authors choose to compare the AamoA/(nirK+nirS) ratio specifically? What is the hypothesis here in the context of N₂O production?

Line 133-135: This statement does not seem accurate in the absolute sense across all sites - looks like there is greater spread in the archaeal amoA numbers, and the mean is higher for arch-amoA. The

change is more pronounced when numbers are normalized to the 16S rRNA abundances, but this is because most archaea in the dataset are AOA whereas most bacteria are not AOB. The 16S rRNA-normalized amoA abundances should not be used as evidence to suggest that AOA outnumber AOB. Again, comparing the absolute qPCR abundances will suffice.

Lines 141-142: Please add references

Lines 140-143: This sentence does not sufficiently explain why the authors expect a correlation between comammox amoA and N₂O emissions.

Lines 143-144: This statement is too general. Please explain which specific genes, and why this is relevant to the discussion.

Lines 144-147: There should be an attempt to discuss these results here since there is no separate Discussion section as in a traditional journal format.

Lines 149-150: what type of data was used to calculate the Shannon diversity index? (i.e., qPCR abundances, metagenome relative abundances or presence-absence?)

Lines 149-150: Please discuss why higher N-cycle functional gene diversity may contribute to N₂O emissions.

152: Non-specific phrasing – ‘N related genes’

Lines 153-154: What about sites missing some of the genes? Please compare and contrast, while discussing the potential mechanisms at play

Lines 165-166: But isn't this expected since most Archaea in the dataset are Thaumarchaeota/AOA? (and AOB are but a small fraction of the bacterial community?)

Lines 167-168: I do not think the data/analysis presented so far unequivocally show that archaea substantially contribute to N₂O production in wetlands (there is no functional analysis here, and this conclusion appears to be drawn solely based on correlations)

Line 178: Are the authors implying that AOA will increase in abundance at higher soil temperatures? Would any effect on N₂O be limited by lower AOB numbers?

Paragraph in lines 183-194: I do not see why this paragraph is necessary for the story. There is a sudden shift in focus to community ecology of AOA versus AOB, which has not been discussed in the context of the thesis of the study (N₂O in wetlands). If the authors choose to retain this paragraph, please relate these observations to N₂O emissions.

Lines 200-201: Please rephrase so that it doesn't sound like this study is trying to provide evidence for the role of archaea in ammonia oxidation.

Lines 243-244: Were the samples homogenized before subsampling?

Lines 330-338: Please provide the qPCR protocol used to amplify each gene (reagent concentrations and cycling conditions)

Figure 2, panels A and B: Please specify how the gene relative abundances were calculated. (i.e., were the gene abundances normalized to the abundances of all predicted genes in each metagenome or to all functional genes excluding rRNAs, or normalized in some other way?). Italicize the gene abbreviations.

Figure 2, panel B: How much of an influence does the largest amoA abundance value (~0.0012) have on the correlation strength? Looks like the largest N₂O numbers are associated with ~0.004-0.007 amoA relative abundances. Are there latitudinal and/or habitat-related patterns here that might be worth pointing out?

Also, how were the error bars on the amoA abundances calculated? Am I mistaken to assume that each point on the plot is derived from a single metagenome? (if that's the case, I'm curious as to how the error bars were derived for amoA abundances)

Line 445: I don't see any error bars in panel C.

Figure 3: I'm not sure how each panel in Fig. 3 supports the figure title 'microbial functional diversity is the main determinant of N₂O production.' Panel C appears to be an attempt to illustrate this, however, only the N cycle functional diversity has been plotted here. Also, the correlation is not very strong. If this is the primary evidence being used to relate functional diversity and N₂O fluxes, I would be a bit skeptical.

Extended data fig. 1: Please add units to the X axes.

Extended data fig. 2: Please add units to each graph. Italicize gene names. Are these qPCR abundances presented in log scale?

Extended data fig. 8: Would this plot be different if the relative abundances of the functional genes are also accounted for? For the blue circles, which genes are generally missing, or is it fairly random?

Reviewer #2 (Remarks to the Author):

The manuscript focuses on the fluxes of N₂O from wetlands around the world, and how these fluxes correlate with different environmental variables and the abundance and structure of the microbial community. The authors logically guide the reader from in-situ measurements of N₂O emission, to the abundance of functional N-cycle genes, and the correlation between these and several environmental factors. Given the high global warming potential of N₂O and – relative to CO₂ and CH₄ – low number of studies, this thorough work is an important contribution to our understanding of N₂O dynamics. I have

therefore very much enjoyed reading and reviewing this manuscript. I would like to ask the authors to clarify a few points related to the methods and (presentation of) the results. I have listed my comments below.

Comments

- Abstract

o P2, L28; The abstract mentions both wetland soils being the greatest source of N₂O, and global warming turning wetland soils from a sink into a source. This sounds somewhat contradictory.

- Main

o P2, L52.; Reference should probably be #2?

o P3, L80; Extended Data Fig 1 could be combined into a single figure by giving a different shape to tropical/non-tropical data points. In the left figure (A), there appear to be one of more datapoints that are blue (near the Y-axis). The figure on the right (B) is not very easy to read, with colours for peat extraction and arable land being very similar. The colours in the legend are clearer than those in the figure itself.

o P. 3 L. 84-85; "N₂O production declines towards higher latitudes in relation to temperature but independently from land-use type" Consider separating these statements (high-latitude and land-use types across temperature), since direct correlations between land-use type and latitude are not presented.

o P4, L90; Could refer to Ext Dat Fig 3A. There is a lot of information packed into this figure, so where possible, please refer to the specific panel.

o P5, L132-134; Archaeal amoA outnumbered bacterial amoA, especially relative abundance, but this difference does not appear to be significant. If it is, please add stats. Also, I was wondering where the statement that archaeal nitrifiers are dominant in the tropics is based on. I believe this is presented in Ext Data Fig 9? If so, please refer to presented data.

o P7, L. 178-179; Could you elaborate on this statement? What constitutes a "high" temperature. Would you expect a tipping point? Can you draw any information of such cut off values from your own data? Many graphs appear to have a value after which emissions increase considerably.

o How does Fig 2B differ from Extended Data Fig. 6 K? Both plot Archaeal amoA abundance against N₂O emission, but they show a different r². Would this only be due to the fact that in Fig 6K, the x-axis has been log-transformed, or are there other differences between the two plots?

o P7, L199; Please argue why the contribution of Archaea to N cycling is especially true in disturbed tropical wetlands. I agree that your data supports the positive relationships between Archaea abundance/diversity (especially Archaeal amoA) or N₂O emission with higher temperature and lower latitudes, which implies that this holds true especially for the tropics. I had, however, difficulty finding a direct relationship with disturbed wetlands, let alone disturbed, tropical wetlands.

- Methods:

o P8-9; The number of sites is a bit unclear, as there are a lot of numbers referring to regions (27), sites (76 \diamond in Figure 1, I only could 74), stations (196), chambers (645), soil samples (628). Please elaborate.

o P. 8, L. 223-228. The methods describe that the sites were characterised according to land use (four grade agricultural intensity index), but this index is not used when presenting the results in the main text.

o P9, L236-237. Gas samples were collected from the headspace of the chambers every 20 mins during 1 hr, which results in a maximum of 4 samples. In the analyses section, however, 3-5 samples per chamber

are mentioned.

o P9, L242. Soil samples were collected from 0-10 cm depth. Please specify if samples were mixed and/or homogenised.

o P9, L242-243. Samples were transported to Estonia. How much time between sampling and storage/analyses? Were they transported cold, or were DNA samples fixed?

o P9, L260. N₂ analyses of intact soil cores. I cannot recall seeing this data mentioned or presented in the main text. Please specify where these results are discussed. What was the basis used for selection of sites (252 out of 645 chambers at 26 out of 74/76 sites)? Temperature was kept similar to field condition during incubation. Does that mean that samples from different climates were run or kept separately?

o P10, L274. Are the methods described for soil analyses correct? In my experience, KCl extractions are used to determine plant-available nitrogen, whereas Olsen- or Bray's method are used for plant-available P. I don't think you can determine plant available K using a KCl extraction method. I assume this plant available K was derived from the NH₄-acetate extraction? Could you provide references for the methods you used or describe them in more detail? For example, the extraction methods and concentrations are not described for all.

o P10, L292-293. Why were 628 soil samples "selected" for DNA analyses instead of all soil samples?

o P10, L288; Apologies if explained, but I could not find whether 0's (or values below detection limit) were included in your analyses.

- Figures

o Fig 2C ◊ what do the blue lines mean? Those are sign?

o Fig3A; I would suggest to remove the correlations between different soil characteristics, and mainly focus on the correlations between environmental parameters on the N₂O emission and abundance of the different functional genes. That way, the figure becomes a lot easier to read. To include correlations between different functional genes, the table could also be cut into two separate, simpler tables.

o Ext Data Fig 1. This figure could also be combined into a single graph, by using different colours and shapes. The different colours in right graph are difficult to discern. In the left graph, there appear to be a few blue dots close to the Y-axis

o Ext Fig3+6+9+10; Please only draw a model when this is significant.

o Ext Fig 8; Please provide statistical information, or, remove figure. It does not add a lot to the overall story, and there are a great many figures in this manuscript.

Reviewer #1 (Remarks to the Author):

This study examines wetland microbial community structure along latitudinal gradients, and relates the abundances of specific taxa and functional genes to nitrous oxide fluxes. Primarily by using correlation analyses, the study posits that archaea play a more important role in wetland N₂O emission than previously recognized. The greatest strength of this study is the sample size as well as the vast geographic coverage enabling comparisons across wetland systems under different habitat and land use conditions. The paper is generally well-written. Methods are generally robust and are explained with sufficient detail. Discussion of the results can often be improved by providing better context (please see specific comments below).

Response: We thank the reviewer for positive assessment of our work and constructive comments.

My main concern with this study is that I do not believe the conclusions as stated are sufficiently supported by the data presented. For example, the two primary takeaways are that: (i) archaea drive N₂O production in wetlands; and (ii) the greater the diversity of N-cycling functional groups, the greater the N₂O fluxes. Since this study design only involved correlation-based analyses, I think the authors should refrain from conclusively assigning causation. I apologize if I missed something, but it appears to me that the above conclusions are derived solely from the fact that N₂O values correlated the best with (i) archaeal abundances and (ii) the Shannon diversity index for N-cycling functional genes. Mechanistic explanations/hypotheses, if any, are not given for either of the two points. The association between AOA and N₂O fluxes is really interesting, and has not been identified prior to this study. However, I don't think there is sufficient evidence to state that AOA drive N₂O fluxes just based on the correlations.

In general, I would like to see better contextualization of the results, acknowledging the limitations of the study design.

Response: We have now revised the entire text to avoid implying causation based on correlations, and discussed our findings in the context of underlying mechanisms. Specific responses are presented below.

Specific comments:

Line 31: Contradicts with the first sentence where wetlands are highlighted as the greatest source of N₂O.

Response: Changed to "a greater source of N₂O".

Line 33-34: A minor but important distinction – the authors examined the functional potential of microbial communities, not the 'gene functions'

Response: Changed to "potential functional".

Line 34: Please include the actual number instead of '>600'

Response: Fixed.

Line 37: I'm not sure what the authors mean by the 'diversity of N-cycle processes driven by nitrifying microbes.' Please note that the term 'nitrifiers' include both ammonia and nitrite oxidizers (I assume the authors are specifically referring to ammonia oxidizers here). Still, these organisms do not mediate a diverse range of N-cycle processes – please be more specific as to

which processes the authors are referring to.

Response: Changed to “functional diversity of microbes”.

Line 39: While I agree N₂O fluxes correlated best with archaeal abundances, I’m not convinced that this is ‘unambiguous evidence’ for a dominant role of archaea in N₂O production. This study has not tested causal links between wetland archaea and N₂O production.

Response: We revised the sentence as: “We further provide evidence that despite their much lower abundance compared to bacteria, nitrifying archaeal abundance is a key factor explaining N₂O emissions from wetland soils globally.”.

Line 52 and throughout: Please check the order of the references. For example, in line 52, the reference should be numbered 2, not 5.

Response: Fixed.

Line 60-61: Please add references.

Response: Added Prosser, JI., et al. *Global Change Biology* 26.1 (2020): 103-118.

Line 68: I would suggest changing ‘N cycling’ to ‘N₂O emissions’

Response: Changed as suggested by the reviewer.

Line 77: Which figure shows the results of testing archaeal diversity as a predictor of N₂O fluxes? Does this refer to AOA diversity alone, or the archaeal phylum? If there is a correlation here, how could it be explained mechanistically?

Response: We removed that statement. The taxonomic diversity *per se* had a weak relation to N₂O fluxes. We have now clarified this alongside a discussion on potential mechanisms on lines 191-207. We additionally performed a genus-level analysis, including all genera uncovered by metagenomes and found only two genera from Thaumarchaeota showed significant correlation with N₂O. We have now added this information and a new Suppl Table 2, please see lines 109-116 & lines 125-131.

Lines 94-99: Please discuss these findings in the context of previous studies.

Response: We have now discussed our results in the context of other studies, please see lines 94-100.

Line 98: The water table depth? Also, looks like this is only true for SAP, and even then, the variation explained is almost close to zero?

Response: Yes we meant the water table depth. It is indeed a weak relationship. We changed the sentence (lines 96-97) to “fungal diversity showed a weak relationship with environmental factors”

Lines 103-104 and throughout: Please always use the complete gene names. This should be 16S and 18S rRNA gene metabarcoding.

Response: Edited throughout the text.

Line 105: I’m not sure if the Extended figure 5 shows this. Is there another figure showing the phyla abundances?

Response: We have now added a new figure (Fig. 2) showing the relative abundance of different phyla based on metagenomics data, which supports our previous statement regarding the most abundant phyla in global wetlands.

Line 107-108: Based on Fig. 2, how are the authors concluding that AOA/Thaumarchaeota are the main predictor of N₂O fluxes? They co-vary, as can be seen in the PLS regression plot, but isn't that correlation only? Did the authors perform a separate prediction task that is not plotted in Fig. 2?

Response: Yes. it is only correlation. We rephrased the statement as “the relative abundance of AOA from the phylum Thaumarchaeota, emerged as the most strongly correlated group with N₂O production (Fig. 3).” We have also provided more context for this finding on lines 109-116.

Line 114: Change ‘mostly costly’ to ‘most costly’

Response: Done.

Lines 114-116: Do the r and p values in parantheses refer to OTU correlations with N₂O fluxes? The placement is a bit confusing.

Response: Yes, we rephrased the statement as follows: “Furthermore, N₂O fluxes showed a strong correlation with the relative abundance of OTUs most closely associated with ‘*Candidatus Nitrosocosmicus oleophilus* MY3’ (Spearman's rank-correlation $r=0.515$, $p<0.001$) and ‘*Candidatus Nitrosotenuis chungbukensis* MY2’ ($r=0.532$, $p<0.001$).”

Lines 126-127: How do the results stated in this paragraph show that nitrification activity is associated with AOA relative abundance since activity was not measured in this study?

Response: Rephrased as follows: “These results add support that the genetic potential for nitrification in these wetland soils is mainly identified by AOA relative abundance.”

Lines 132-133: Figure 1 does not provide evidence for archaeal dominance in the tropics. It would be really helpful to have the raw qPCR abundances for these genes since ratios can obscure meaning. For instance, here the *AamoA*/(*nirK*+*nirS*) ratio could be higher in the tropics either due to an increase in AOA numbers or due to a lower abundances of *nirK* or *nirS*-harboring denitrifiers.

Response: We have now added two new subfigures (Fig. 1c,d) to Fig. 1, showing the distribution of *AamoA* and *nir* across latitude. As shown in the new figure, *AamoA* shows a significant increase towards tropical regions in contrast to *nir*. We clarified it accordingly: “The relative increase in archaeal nitrifiers compared to denitrifiers ...”. In addition, the relationship between N₂O and environmental variables with the raw qPCR abundances have now been presented in the Suppl. Fig. 4.

Lines 132-133 and Figure 1: Why did the authors choose to compare the *AamoA*/(*nirK*+*nirS*) ratio specifically? What is the hypothesis here in the context of N₂O production?

Response: Nitrification and denitrification processes are shown globally to be the main processes driving the N₂O emissions from different environments. In the wetlands, it was previously thought that denitrification was the main driver of the N₂O emissions, but we were able to show the importance of archaeal nitrification, especially in the context of producing N₂O. As genes *nirK* and *nirS* are the main indicators for describing the N₂O emission from the denitrification process, we used the ratio *AamoA*/(*nirK*+*nirS*) to describe the balance between the processes, which are the primary sources of N₂O. We rephrase the hypothesis as follows: “We hypothesized that the high N₂O production in certain global wetland soils is mainly related to the diversity and abundance of nitrifying microbes, and that archaeal nitrifiers, both at absolute abundance and in ratio to denitrifiers, are the most robust and accurate explanatory factor of N₂O emissions from wetland soils globally.”

Line 133-135: This statement does not seem accurate in the absolute sense across all sites - looks like there is greater spread in the archaeal *amoA* numbers, and the mean is higher for arch-*amoA*.

The change is more pronounced when numbers are normalized to the 16S rRNA abundances, but this is because most archaea in the dataset are AOA whereas most bacteria are not AOB. The 16S rRNA-normalized *amoA* abundances should not be used as evidence to suggest that AOA outnumber AOB. Again, comparing the absolute qPCR abundances will suffice.

Response: Yes, that may cause biases. We have now rephrased our statement and removed the 16S rRNA-normalized *amoA* abundances and the subplot B of Suppl. Fig. 8.

Lines 141-142: Please add references

Response: Added two references for the statement.

Lines 140-143: This sentence does not sufficiently explain why the authors expect a correlation between comammox *amoA* and N₂O emissions.

Response: Indeed we expected a weak correlation. We edited the statement. Please see lines 167-171.

Lines 143-144: This statement is too general. Please explain which specific genes, and why this is relevant to the discussion.

Response: We specified the statement as follows: "The reads related to anaerobic ammonium oxidation (anammox) and the nitrite/nitrate-dependent anaerobic methane oxidation (n-damo) did not correlate with the N₂O fluxes."

Lines 144-147: There should be an attempt to discuss these results here since there is no separate Discussion section as in a traditional journal format.

Response: We added a more detailed discussion on lines 179-190.

Lines 149-150: what type of data was used to calculate the Shannon diversity index? (i.e., qPCR abundances, metagenome relative abundances or presence-absence?)

Response: The abundance quantified by qPCR, we clarified this. as "we related N₂O emissions with the diversity of all major genes involved in the N cycle (based on their absolute abundances quantified by qPCR)"

Lines 149-150: Please discuss why higher N-cycle functional gene diversity may contribute to N₂O emissions.

Response: We added a discussion on lines 193-204.

152: Non-specific phrasing – 'N related genes'

Response: We changed it: "N cycle related genes".

Lines 153-154: What about sites missing some of the genes? Please compare and contrast, while discussing the potential mechanisms at play

Response: Over all samples, 9% of genes were absent from tropical sites compared to 15% from the non-tropical sites, and no big contrast between lacking different functional genes. However, as the other reviewer suggested, we removed the Extended Data Fig. 8. In addition, we rewrote the section on lines 191-208,

Lines 165-166: But isn't this expected since most Archaea in the dataset are Thaumarchaeota/AOA? (and AOB are but a small fraction of the bacterial community?)

Response: Euryarchaeota was relatively more abundant than Thaumarchaeota in our metagenomes, but the former showed no significant relationship with N₂O in contrast to the latter. In addition, our new genus-level analysis showed that two genera with relatively average

abundant archaeal genera show the strongest correlation with N₂O fluxes (please see Suppl. Table 2).

Lines 167-168: I do not think the data/analysis presented so far unequivocally show that archaea substantially contribute to N₂O production in wetlands (there is no functional analysis here, and this conclusion appears to be drawn solely based on correlations)

Response: We rephrased this as: “These results further support the potential key role of nitrifying archaea in N₂O emissions from wetland soils globally.”

Line 178: Are the authors implying that AOA will increase in abundance at higher soil temperatures? Would any effect on N₂O be limited by lower AOB numbers?

Response: We rephrased the statement as: “This finding suggests that high temperature (>20 °C) in combination with optimal soil moisture¹³ may promote N₂O emissions from soils due to an increased AOA abundance. Nevertheless, how this may be offset by their altered balance with AOB remains to be seen, as especially AOB seem to be more adapted to N₂O consumption below 20 °C¹³.”

Paragraph in lines 183-194: I do not see why this paragraph is necessary for the story. There is a sudden shift in focus to community ecology of AOA versus AOB, which has not been discussed in the context of the thesis of the study (N₂O in wetlands). If the authors choose to retain this paragraph, please relate these observations to N₂O emissions.

Response: We agree with the reviewer and deleted the section.

Lines 200-201: Please rephrase so that it doesn't sound like this study is trying to provide evidence for the role of archaea in ammonia oxidation.

Response: Agreed. We changed the statement to “Furthermore, we provide evidence that archaeal abundance is a key factor associated with N₂O production in ammonia oxidation in wetland soils globally.”

Lines 243-244: Were the samples homogenized before subsampling?

Response: Yes, they were. We changed it: “The homogenized samples were divided into subsamples for physical-chemical analyses and DNA extraction.”

Lines 330-338: Please provide the qPCR protocol used to amplify each gene (reagent concentrations and cycling conditions)

Response: We clarified it in the section “Quantitative PCR” and added Supplementary Table 9.

Figure 2, panels A and B: Please specify how the gene relative abundances were calculated. (i.e., were the gene abundances normalized to the abundances of all predicted genes in each metagenome or to all functional genes excluding rRNAs, or normalized in some other way?). Italicize the gene abbreviations.

Response: This information was already present in the Methods, but we have now added it to the figure legend as well: “The gene relative abundances in **a** and **b** were normalized by the total number of quality filtered metagenomics reads.”

Figure 2, panel B: How much of an influence does the largest *amoA* abundance value (~0.0012) have on the correlation strength? Looks like the largest N₂O numbers are associated with ~0.004-0.007 *amoA* relative abundances. Are there latitudinal and/or habitat-related patterns here that might be worth pointing out?

Response: Removing the largest *amoA* abundance value had little effect on the correlation strength (see Fig. 1 below). Indeed when we separated the sites with relative abundance of AOA falling into 0.004-0.007, these sites were significantly more presented in low latitude sites (F-

statistic: 9.745 on 1 and 72 DF, p-value: 0.002588; see Fig. 2 below). Archaeal *amoA* showed a positive relationship to max temperature of the warmest month ($r^2=0.180$, $p<0.001$).

Figure 1.

Figure 2.

Also, how were the error bars on the *amoA* abundances calculated? Am I mistaken to assume that each point on the plot is derived from a single metagenome? (if that's the case, I'm curious as to how the error bars were derived for *amoA* abundances)

Response: Each point represents a site mean. We clarified this in the figure legend: "Relationship between site mean relative abundance of archaeal *amoA* and N₂O production. Taxa and gene

relative abundances were quantified using metagenomics. Error bars represent standard errors (SE) of the site means.”

Line 445: I don't see any error bars in panel C.

Response: Our apologies, error bars referred to panel B. We have now edited the legend.

Figure 3: I'm not sure how each panel in Fig. 3 supports the figure title 'microbial functional diversity is the main determinant of N₂O production.' Panel C appears to be an attempt to illustrate this, however, only the N cycle functional diversity has been plotted here. Also, the correlation is not very strong. If this is the primary evidence being used to relate functional diversity and N₂O fluxes, I would a be bit skeptical.

Response: We edited the legend as “Nitrogen-cycle genes as the main factors explaining N₂O emissions across the global wetland soils.” Indeed, the correlation is modest owing to large variability across the globe. However, the trend is evident and significant.

Extended data fig. 1: Please add units to the X axes.

Response: Done.

Extended data fig. 2: Please add units to each graph. Italicize gene names. Are these qPCR abundances presented in log scale?

Response: Edited accordingly. Yes, all values on y-axis are in log scale - we added this information to the figure legend.

Extended data fig. 8: Would this plot be different if the relative abundances of the functional genes are also accounted for? For the blue circles, which genes are generally missing, or is it fairly random?

Response: We have removed this figure in the revised manuscript as suggested by Reviewer #2.

Reviewer #2 (Remarks to the Author):

The manuscript focuses on the fluxes of N₂O from wetlands around the world, and how these fluxes correlate with different environmental variables and the abundance and structure of the microbial community. The authors logically guide the reader from in-situ measurements of N₂O emission, to the abundance of functional N-cycle genes, and the correlation between these and several environmental factors. Given the high global warming potential of N₂O and – relative to CO₂ and CH₄ – low number of studies, this thorough work is an important contribution to our understanding of N₂O dynamics. I have therefore very much enjoyed reading and reviewing this manuscript. I would like to ask the authors to clarify a few points related to the methods and (presentation of) the results. I have listed my comments below.

Response: We thank the reviewer for the positive feedback and constructive comments.

Comments

- Abstract

o P2, L28; The abstract mentions both wetland soils being the greatest source of N₂O, and global warming turning wetland soils from a sink into a source. This sounds somewhat contradictory.

Response: Changed the last sentence of the abstract as follows: “transforming wetland soils to a greater source of N₂O.”

- Main

o P2, L52.; Reference should probably be #2?

Response: fixed.

o P3, L80; Extended Data Fig 1 could be combined into a single figure by giving a different shape to tropical/non-tropical data points. In the left figure (A), there appear to be one of more datapoints that are blue (near the Y-axis). The figure on the right (B) is not very easy to read, with colours for peat extraction and arable land being very similar. The colours in the legend are clearer than those in the figure itself.

Response: We have revised the figure accordingly.

o P. 3 L. 84-85; “N₂O production declines towards higher latitudes in relation to temperature but independently from land-use type” Consider separating these statements (high-latitude and land-use types across temperature), since direct correlations between land-use type and latitude are not presented.

Response: We rephrased the statement as follows: “Assessing environmental determinants of N₂O fluxes revealed that N₂O emissions decline towards higher latitudes (Fig. 1, Supplementary Fig. 1).”

o P4, L90; Could refer to Ext Dat Fig 3A. There is a lot of information packed into this figure, so where possible, please refer to the specific panel.

Response: Done.

o P5, L132-134; Archaeal *amoA* outnumbered bacterial *amoA*, especially relative abundance, but this difference does not appear to be significant. If it is, please add stats.

Response: We rephrased as follows: “The absolute archaeal *amoA* abundance was slightly higher than the bacterial *amoA* abundance (qPCR: F= 6.00, p=0.015), substantiating the importance of archaea in nitrification across wetland soils (Supplementary Fig. 7), as previously reported for grassland and agricultural soils²⁰.”

Also, I was wondering where the statement that archaeal nitrifiers are dominant in the tropics is based on. I believe this is presented in Ext Data Fig 9? If so, please refer to presented data.

We have now rephrased our statement and added new figures (Fig. 1b,c): “The absolute archaeal *amoA* abundance was slightly higher than the bacterial *amoA* abundance (qPCR: F= 6.00, p=0.015), substantiating the importance of archaea in nitrification across wetland soils (Supplementary Fig. 7), as previously reported for grassland and agricultural soils²⁰.”

o P7, L. 178-179; Could you elaborate on this statement? What constitutes a “high” temperature. Would you expect a tipping point? Can you draw any information of such cut off values from your own data? Many graphs appear to have a value after which emissions increase considerably.

Response: We rephrased as follows: “This finding suggests that high temperature (>20 °C) in combination with optimal soil moisture¹³ may promote N₂O emissions from soils due to an increased AOA abundance. Nevertheless, how this may be offset by their altered balance with AOB remains to be seen, as especially AOB seem to be more adapted to N₂O consumption below 20 °C¹³.”

No remarkable temperature tipping points were found. Moreover, the effect of soil temperature is strongly intertwined with the effect of soil moisture.

o How does Fig 2B differ from Extended Data Fig. 6 K? Both plot Archaeal *amoA* abundance against N₂O emission, but they show a different r². Would this only be due to the fact that in Fig 6K, the x-axis has been log-transformed, or are there other differences between the two plots?

Response: Please note Fig 2b is based on the metagenomics relative abundance of Archaeal *amoA* while Extended Data Fig. 7k (now Supplementary Fig. 6k) shows the absolute abundance of

Archaeal *amoA* abundance quantified by qPCR. The reason we present both figures is that they were quantified by two independent methods and two different measures (absolute and relative abundance). We have now clarified where we used qPCR or metagenomics data in all figure legends.

o P7, L199; Please argue why the contribution of Archaea to N cycling is especially true in disturbed tropical wetlands. I agree that your data supports the positive relationships between Archaea abundance/diversity (especially Archaeal *amoA*) or N₂O emission with higher temperature and lower latitudes, which implies that this holds true especially for the tropics. I had, however, difficulty finding a direct relationship with disturbed wetlands, let alone disturbed, tropical wetlands.

Response: We rephrased this section as: “Considering the combined effect of optimal soil moisture and temperature, archaea are important contributors to N cycling in the wetland soils, particularly in drained tropical wetland soils⁶..”

- Methods:

o P8-9; The number of sites is a bit unclear, as there are a lot of numbers referring to regions (27), sites (76 \diamond in Figure 1, I only could 74), stations (196), chambers (645), soil samples (628). Please elaborate.

Response: We double checked the numbers and corrected them. The correct numbers are 29 regions, 76 sites, 197 stations, 645 chambers and 645 soil samples. Sorry about this discrepancy

o P. 8, L. 223-228. The methods describe that the sites were characterised according to land use (four grade agricultural intensity index), but this index is not used when presenting the results in the main text.

Response: Apologies, it should have been “a four-grade land-use intensity index”, which is used in Supplementary Fig. 1. We edited the methods.

o P9, L236-237. Gas samples were collected from the headspace of the chambers every 20 mins during 1 hr, which results in a maximum of 4 samples. In the analyses section, however, 3-5 samples per chamber are mentioned.

Response: We collected 4 samples an hour, except the first sampled region (Iceland in 2011) where we collected 5 samples per hour. After that we optimized the number of samples to 4 per hour. During the quality control, in some cases we removed one outlier. Therefore, we remain with the original text “Gas samples were collected from the headspace of the chambers every 20 mins during 1 hr”. We clarified the section on lines 304-307: “During the quality control, in cases of insignificant regression ($p > 0.05$) we removed one outlier. If the regression remained insignificant but the flux value fell below the gas-chromatography measuring accuracy (regression change of N₂O concentration $\delta v < 10$ ppb) we included it in the subsequent analyses as a zero value.”

o P9, L242. Soil samples were collected from 0-10 cm depth. Please specify if samples were mixed and/or homogenised.

Response: We clarified the statement as: “The homogenized samples were divided into subsamples for physical-chemical analyses and DNA extraction.”

o P9, L242-243. Samples were transported to Estonia. How much time between sampling and storage/analyses? Were they transported cold, or were DNA samples fixed?

Response: Transport time was 1–3 days. As we stated below: “During transport, the soil samples were kept below the ambient soil temperature from which they were collected.”

o P9, L260. N₂ analyses of intact soil cores. I cannot recall seeing this data mentioned or presented in the main text. Please specify where these results are discussed.

Response: We added relevant discussion on lines 185-188: “Finally, consumption of N₂O by *nosZ* complicated the effect of denitrification gene abundances. This was shown by the potential N₂ flux peaking in the temperate climate in negative correlation with soil NO₃ content and agricultural intensity, and in positive correlation with soil water content (Fig. 4a; Supplementary Fig. 9).”.

What was the basis used for selection of sites (252 out of 645 chambers at 26 out of 74/76 sites)? Temperature was kept similar to field condition during incubation. Does that mean that samples from different climates were run or kept separately?

Response: The period of investigation lasted from 2011–2018. N₂ analyses were made for the samples since 2014. Samples from different climates were run at respective temperatures. We have now clarified this in the methods.

o P10, L274. Are the methods described for soil analyses correct? In my experience, KCl extractions are used to determine plant-available nitrogen, whereas Olsen- or Bray’s method are used for plant-available P. I don’t think you can determine plant available K using a KCl extraction method. I assume this plant available K was derived from the NH₄-acetate extraction? Could you provide references for the methods you used or describe them in more detail? For example, the extraction methods and concentrations are not described for all.

Response: There was a mistake in our description. Plant-available phosphorus (P, NH₄-lactate extractable) was determined on a FiaStar5000 flow-injection analyzer. The detailed protocol of the soil physical and chemical analyses is described in Pärn et al. 2018 (Nature Communications). We have now edited this section on lines 323-324 & 334-335.

o P10, L292-293. Why were 628 soil samples “selected” for DNA analyses instead of all soil samples?

Response: Our apologies, the correct number is 645 as all soil samples were used for the DNA analyses. We corrected the number in the text.

o P10, L288; Apologies if explained, but I could not find whether 0’s (or values below detection limit) were included in your analyses.

Response: Indeed, we included the zeros. We clarified it on lines 304-307 as: “During the quality control, in cases of insignificant regression ($p > 0.05$) we removed one outlier. If the regression remained insignificant but the flux value fell below the gas-chromatography measuring accuracy (regression change of N₂O concentration $\delta v < 10$ ppb) we included it in the subsequent analyses as a zero value.”

- Figures

o Fig 2C ◊ what do the blue lines mean? Those are sign?

Response: Blue lines represent archaeal phyla. We have added this to the legend.

o Fig3A; I would suggest to remove the correlations between different soil characteristics, and mainly focus on the correlations between environmental parameters on the N₂O emission and abundance of the different functional genes. That way, the figure becomes a lot easier to read. To

include correlations between different functional genes, the table could also be cut into two separate, simpler tables.

Response: We have now simplified the figure by removing soil variables showing weak relationship with N₂O emissions or the functional genes from the figure. We kept other variables in the figure because we intend to give an overall overview of the relationship between genes and environmental variables and gas fluxes.

o Ext Data Fig 1. This figure could also be combined into a single graph, by using different colours and shapes. The different colours in right graph are difficult to discern. In the left graph, there appear to be a few blue dots close to the Y-axis

Response: Agreed. We have merged the figures into one.

o Ext Fig3+6+9+10; Please only draw a model when this is significant.

Response: Done.

o Ext Fig 8; Please provide statistical information, or, remove figure. It does not add a lot to the overall story, and there are a great many figures in this manuscript.

Response: We removed Ext Fig 8 from the revised manuscript.

REVIEWER COMMENTS

Reviewer #1 (Remarks to the Author):

General comments to the authors:

I thank the authors for carefully addressing the comments and revising the manuscript accordingly. I particularly appreciate the additional effort put into discussing the results in the context of prior work.

Specific comments:

Line 35: Change “due to” to “and correlate with”

Line 58-59: Minor point, but a slight re-arrangement of the sentence will really enhance clarity here. Currently, it could mean AOA are providing nitrite/nitrate ‘directly and indirectly’ to denitrifiers, whereas I think the authors mean AOA are directly producing N₂O while also indirectly affecting N₂O production by providing substrates for denitrification.

Line 61-62: Not sure if I agree. Also, I don’t think this sentence is needed here anyway.

Line 62: I believe the authors mean a stoichiometric difference in N₂O production between AOA and AOB, not a difference in the absolute sense (i.e., AOA produce less N₂O per nitrite produced than AOB?).

Line 75: Just a suggestion for re-phrasing: “archaeal nitrifiers, both in terms of absolute abundance and their relative abundance with respect to denitrifiers, ...”

Line 111-112: Not true! There are many lineages of Thaumarchaeota that do not oxidize ammonia (in addition to the 1.1c group highlighted here). Please see:

1. Ren et al. 2019. Phylogenomics suggests oxygen availability as a driving force in Thaumarchaeota evolution. *ISME J.* 13(9):2150-2161. doi: 10.1038/s41396-019-0418-8.
2. Reji L, Francis CA. 2020. Metagenome-assembled genomes reveal unique metabolic adaptations of a basal marine Thaumarchaeota lineage. *ISME J.* 14(8):2105-2115. doi: 10.1038/s41396-020-0675-6.
3. Aylward FO, Santoro AE. 2020. Heterotrophic Thaumarchaea with Small Genomes Are Widespread in the Dark Ocean. *mSystems.* 5(3):e00415-20. doi: 10.1128/mSystems.00415-20.

Line 116: Cut ‘of’ after total

Line 118: Please italicize taxonomic names throughout. The phylum names don’t have to be italicized.

Lines 132-133: OGs usually refer to “(clusters of) orthologous groups.” I couldn’t find any information in the Methods on how OGs were inferred. Based on the OG annotation used for amoA in line 134, I believe this was obtained via eggnog-mapper, but please clarify this in Methods.

Line 148: I do not follow this argument. Aerobic CO dehydrogenases are found in carboxydrotrophic

bacteria. Not sure why they would be active under flooded conditions when the system is most likely anoxic. Similarly, pyruvate dehydrogenase is typically used under aerobic conditions (and is replaced with pyruvate:ferredoxin oxidoreductase under anaerobic conditions). The cited paper is too generic – it does not support the specific idea proposed here.

Lines 144-148: These statements seem ill-placed here as they are unrelated to the AOA story being discussed.

Line 157 and throughout: Abbreviations of gene names are italicized, however, 'AamoA' here is an abbreviation that the authors have come up with to refer to archaeal ammonia oxygenase, and is not a conventional gene name. Therefore, I suggest either not using italics or replacing this with 'arch-amoA' and 'bac-amoA'

Line 179: Why are the denitrification gene inactive? Were the samples fully oxic? Denitrification can occur in anoxic microenvironments even within otherwise aerobic systems

Line 185-186: I do not follow the meaning of this sentence.

Line 202: change "contributed the most to" to "correlated the best with"

Line 203-204: The phrase after the comma seem unrelated to the first half of the sentence

Lines 209-210: suggestion for rephrasing: "we compared the nucleotide sequences of the archaeal OTUs correlating with N₂O production with those showing no such correlation"

Line 211: What does 'the genomes' refer to? (based on the details provided in Methods, I believe this is complete archaeal genomes downloaded from GenBank, but worth specifying here)

Line 212-214: Yes, but this is expected because the authors are starting out the search with AOA OTUs.

Lines 212-218: These lines need to be re-written to avoid circular reasoning because as stated in Lines 116-125, the OTUs correlating the best with N₂O fluxes were that of AOA. As a consequence, when blasted against genome databases, the top hits are going to be AOA genomes. This is why the selected genomes are more enriched in amoA, as would be expected.

Line 229: The 'compared with AOB' part is not entirely accurate. The cited study observed specific AOB lineages enriched in the high temperature treatment alongside AOA. Just that AOA were only enriched at high temperatures.

Line 233: What does 'adapted to N₂O consumption' mean? That AOB consume N₂O below 20C? I didn't see evidence for this in the cited paper – apologies if I missed something.

Lines 233-236: I do not believe this is the correct reference. Also, inadequate citations throughout the sentence - please add references for the expected outcomes listed. Do the authors mean comammox

when they say 'complete nitrification'? Are they suggesting that complete nitrification (either via comammox or via a tight coupling of ammonia oxidation and nitrite oxidation) lowers N₂O fluxes? I'm not following this logic, and would really appreciate references.

Line 344-346: Sorry if missed this, but which of the two 16S rRNA gene libraries did the authors ended up using for their downstream analyses? Were the libraries merged (if so, how were relative abundances accounted for?)?

Reviewer #2 (Remarks to the Author):

In this resubmitted manuscript, the authors have responded to the comments raised by myself and the other reviewer in a satisfactory manner. The questions I had regarding the method descriptions have all been answered or corrected in the resubmitted manuscript. I have a few remaining points, which I have listed below.

- While I agree that there is an increase in N₂O emission with temperature, supplementary figure 1 doesn't suggest that this is related to tropical regions (higher N₂O emission rates at 25C in non-tropical regions is comparable to N₂O emission at 30C in tropical regions).

- In the manuscript, you both refer to N₂O production and N₂O emission. Most of this data is based on N₂O fluxes measured in-situ, which means they are a product of N₂O production and consumption (through denitrification). It is therefore better to speak of N₂O emission or flux.

- I was wondering if organisms have a single copy of the amoA gene, or if they can contain multiple copies. Similarly, this question holds for nirK/nirS. This is important when comparing Archaea amoA with Bacterial amoA, or when using ratios of nitrification/denitrification genes.

L. 79; Replace the first "explained" in this sentence

L. 147; First use of formula NO₃  use nitrate (NO₃) instead

L.225-227; I'm still not entire sure what the difference is between these two correlations of amoA with temperature (other than one being linear and one exponential)

L. 257-259; This final sentence remains a bit vague. What kind of changes would be referred to? How would this study be a foundation. I am not sure if such a generic final sentence is really necessary. The previous sentence may be enough as a closing statement. Otherwise, please be more specific.

Reviewer #1 (Remarks to the Author):

General comments to the authors:

I thank the authors for carefully addressing the comments and revising the manuscript accordingly. I particularly appreciate the additional effort put into discussing the results in the context of prior work.

Response: We thank the reviewer for the positive evaluation of our work and constructive comments.

Specific comments:

Line 35: Change “due to” to “and correlate with”

Response: Done.

Line 58-59: Minor point, but a slight re-arrangement of the sentence will really enhance clarity here. Currently, it could mean AOA are providing nitrite/nitrate ‘directly and indirectly’ to denitrifiers, whereas I think the authors mean AOA are directly producing N₂O while also indirectly affecting N₂O production by providing substrates for denitrification.

Response: We changed the sentence (line 57) to “AOA not only directly produce N₂O, but also indirectly affect N₂O production by providing substrates for denitrification.”

Line 61-62: Not sure if I agree. Also, I don’t think this sentence is needed here anyway.

Response: We removed the sentence.

Line 62: I believe the authors mean a stoichiometric difference in N₂O production between AOA and AOB, not a difference in the absolute sense (i.e., AOA produce less N₂O per nitrite produced than AOB?).

Response: We revised the sentence (line 59) as “AOA may play a pivotal, underexplored role in fueling denitrification and facilitating terrestrial N₂O emissions⁸ in many soil environments.”

Line 75: Just a suggestion for re-phrasing: “archaeal nitrifiers, both in terms of absolute abundance and their relative abundance with respect to denitrifiers, ...”

Response: Revised accordingly.

Line 111-112: Not true! There are many lineages of Thaumarchaeota that do not oxidize ammonia (in addition to the 1.1c group highlighted here). Please see:

1. Ren et al. 2019. Phylogenomics suggests oxygen availability as a driving force in Thaumarchaeota evolution. ISME J. 13(9):2150-2161. doi: 10.1038/s41396-019-0418-8.
2. Reji L, Francis CA. 2020. Metagenome-assembled genomes reveal unique metabolic adaptations of a basal marine Thaumarchaeota lineage. ISME J. 14(8):2105-2115. doi: 10.1038/s41396-020-0675-6.
3. Aylward FO, Santoro AE. 2020. Heterotrophic Thaumarchaea with Small Genomes Are Widespread in the Dark Ocean. mSystems. 5(3):e00415-20. doi: 10.1128/mSystems.00415-20.

Response: We thank the reviewer for pointing this out. We removed the statement and wrote the following sentence (line 109) as “A previous study also reports a strong association between the thaumarchaeal 16S rRNA and *amoA* genes in environmental samples¹⁵.”

Line 116: Cut 'of' after total

Response: Done.

Line 118: Please italicize taxonomic names throughout. The phylum names don't have to be italicized.

Response: Done.

Lines 132-133: OGs usually refer to "(clusters of) orthologous groups." I couldn't find any information in the Methods on how OGs were inferred. Based on the OG annotation used for amoA in line 134, I believe this was obtained via egg-nog-mapper, but please clarify this in Methods.

Response: We edited the sentence and added the following to the methods (lines 417-419): "Using this method, we calculated the relative abundance of (clusters of) orthologous groups (OG) by mapping quality-filtered reads against the egg-nog database"

Line 148: I do not follow this argument. Aerobic CO dehydrogenases are found in carboxydrotrophic bacteria. Not sure why they would be active under flooded conditions when the system is most likely anoxic. Similarly, pyruvate dehydrogenase is typically used under aerobic conditions (and is replaced with pyruvate:ferredoxin oxidoreductase under anaerobic conditions). The cited paper is too generic – it does not support the specific idea proposed here.

Lines 144-148: These statements seem ill-placed here as they are unrelated to the AOA story being discussed.

Response: We removed this section from the revised manuscript.

Line 157 and throughout: Abbreviations of gene names are italicized, however, 'AamoA' here is an abbreviation that the authors have come up with to refer to archaeal ammonia oxygenase, and is not a conventional gene name. Therefore, I suggest either not using italics or replacing this with 'arch-amoA' and 'bac-amoA'

Response: We have now used 'archaeal *amoA*' and 'bacterial *amoA*' throughout the text.

Line 179: Why are the denitrification gene inactive? Were the samples fully oxic? Denitrification can occur in anoxic microenvironments even within otherwise aerobic systems

Response: We clarified the issue on lines 183-186 as follows: "The denitrification genes responsible for N₂O production (*nirK* and *nirS*) showed weak or no correlation with N₂O emission (Supplementary Fig. 7). The abundance of *nir* genes was strongly correlated with that of *nosZ* genes (Fig. 4a) that reduce N₂O into inert N₂. This consumption may explain low N₂O emissions from the soils enriched with *nir* genes. Potential N₂ production, however, was not significantly correlated with *nosZ* abundance (Fig. 4a)."

Line 185-186: I do not follow the meaning of this sentence.

Response: We revised the sentence on lines 183-186 – please see our response to the previous comment.

Line 202: change "contributed the most to" to "correlated the best with"

Response: Done.

Line 203-204: The phrase after the comma seem unrelated to the first half of the sentence

Response: We revised the sentence (lines 206-207) as “Among the studied functional genes, the abundance of archaeal *amoA* correlated the best to temperature and C/N ratio (Supplementary Figs. 5), similarly to the N-cycle gene diversity.”

Lines 209-210: suggestion for rephrasing: "we compared the nucleotide sequences of the archaeal OTUs correlating with N₂O production with those showing no such correlation"

Response: Revised accordingly.

Line 211: What does ‘the genomes’ refer to? (based on the details provided in Methods, I believe this is complete archaeal genomes downloaded from GenBank, but worth specifying here)

Response: Yes. We revised the sentence as “Using BlastN searches of 16S rRNA gene reads against complete archaeal genomes.”

Line 212-214: Yes, but this is expected because the authors are starting out the search with AOA OTUs.

Response: We agree. We removed that statement.

Lines 212-218: These lines need to be re-written to avoid circular reasoning because as stated in Lines 116-125, the OTUs correlating the best with N₂O fluxes were that of AOA. As a consequence, when blasted against genome databases, the top hits are going to be AOA genomes. This is why the selected genomes are more enriched in *amoA*, as would be expected.

Response: We rewrote the paragraph on lines 134-141: “To further evaluate the genetic basis facilitating N₂O emission, we compared the nucleotide sequences of the archaeal OTUs correlating with N₂O emission with those showing no such correlation. Using BlastN searches of 16S rRNA gene reads against complete archaeal genomes, we located the closest genome-sequenced relatives and obtained the corresponding genomic functional profiles. Based on these, we found that the aerobic ammonia oxidation pathway was restricted to four archaeal genera belonging to Thaumarchaeota - *Nitrososphaera*, *Nitrosocosmicus*, *Nitrosotenuis*, and *Nitrosarchaeum* (Supplementary Tables 5,6).”

Line 229: The 'compared with AOB' part is not entirely accurate. The cited study observed specific AOB lineages enriched in the high temperature treatment alongside AOA. Just that AOA were only enriched at high temperatures.

Response: We removed “compared with AOB”.

Line 233: What does ‘adapted to N₂O consumption’ mean? That AOB consume N₂O below 20°C? I didn’t see evidence for this in the cited paper – apologies if I missed something.

Lines 233-236: I do not believe this is the correct reference. Also, inadequate citations throughout the sentence - please add references for the expected outcomes listed. Do the authors mean comammox when they say 'complete nitrification'? Are they suggesting that complete nitrification (either via comammox or via a tight coupling of ammonia oxidation and nitrite oxidation) lowers N₂O fluxes? I'm not following this logic, and would really appreciate references.

Response: We rewrote the section as follows (lines 304-313): “We explored the environmental conditions favoring microbial taxa and genes driving N₂O emission. The archaeal *amoA* displayed a unimodal relationship with mean annual air temperature peaking between 15 and 20 °C ($r^2_{adj}=0.233$, $p<0.001$; Supplementary Fig. 9). This supports earlier

findings of greater AOA activities in warmer seasons³². The strong positive correlations of the AOA/AOB ratio, mean annual air temperature and soil temperature (Supplementary Fig. 11) is in line with the relatively high temperature optimum of AOA³³. This finding suggests that elevated (>15 °C) soil temperature in combination with optimal soil moisture³¹ may promote N₂O emissions from soils due to an increased AOA abundance. Nevertheless, how this may be offset by their altered balance with AOB remains to be determined. “

Line 344-346: Sorry if missed this, but which of the two 16S rRNA gene libraries did the authors ended up using for their downstream analyses? Were the libraries merged (if so, how were relative abundances accounted for?)?

Response: We mainly used Illumina dataset for downstream community analysis but also used the long sequences of PacBio representative OTUs for identifying N₂O taxa and closely related genomes. We have now clearly specified where each dataset has been used in the legends of our figures and tables.

Reviewer #2 (Remarks to the Author):

In this resubmitted manuscript, the authors have responded to the comments raised by myself and the other reviewer in a satisfactory manner. The questions I had regarding the method descriptions have all been answered or corrected in the resubmitted manuscript. I have a few remaining points, which I have listed below.

Response: We thank the reviewer for positive assessment of our revision and constructive comments.

- While I agree that there is an increase in N₂O emission with temperature, supplementary figure 1 doesn't suggest that this is related to tropical regions (higher N₂O emission rates at 25C in non-tropical regions is comparable to N₂O emission at 30C in tropical regions).

Response: We revised the manuscript to avoid implying that tropical regions have higher N₂O emission rates, including on lines 304-313.

- In the manuscript, you both refer to N₂O production and N₂O emission. Most of this data is based on N₂O fluxes measured in-situ, which means they are a product of N₂O production and consumption (through denitrification). It is therefore better to speak of N₂O emission or flux.

Response: We changed N₂O production to N₂O emission throughout the manuscript.

- I was wondering if organisms have a single copy of the *amoA* gene, or if they can contain multiple copies. Similarly, this question holds for *nirK/nirS*. This is important when comparing Archaea *amoA* with Bacterial *amoA*, or when using ratios of nitrification/denitrification genes.

Response: To our knowledge, except only one report of a second copy of *amoA* in an AOA genome, all reported AOA have a single copy of *amoA* (Herbold et al. 2017). The number of copies of the *amoA* gene is typically two or three according to the AOB species (Norton et al., 1996; Hommes et al., 1998; Stein et al., 2007). Most of the *nirK*-type denitrifiers have also been found to contain one *nirK* gene copy in their genome, but some of them can contain two and a few of them can contain three *nirK* copies (Helen et al., 2016; Lund et al., 2012). Also, *nirS*-type denitrifiers have been found to contain one or two *nirS* gene copies in their genome (Helen et al., 2016). Therefore, the gene copy numbers are comparable regarding

AOA and both types of denitrifiers, because they are mainly possessing a single copy gene per cell.

Herbold, C. W., Lehtovirta-Morley, L. E., Jung, M. Y., Jehmlich, N., Hausmann, B., Han, P., ... & Gubry-Rangin, C. (2017). Ammonia-oxidising archaea living at low pH: insights from comparative genomics. *Environmental microbiology*, 19(12), 4939-4952.

Helen, D., Kim, H., Tytgat, B., & Anne, W. (2016). Highly diverse nirK genes comprise two major clades that harbour ammonium-producing denitrifiers. *BMC genomics*, 17(1), 1-13.

Hommers, N. G., Sayavedra-Soto, L. A., & Arp, D. J. (1998). Mutagenesis and expression of amo, which codes for ammonia monooxygenase in *Nitrosomonas europaea*. *Journal of Bacteriology*, 180(13), 3353-3359.

Lund, M. B., Smith, J. M., & Francis, C. A. (2012). Diversity, abundance and expression of nitrite reductase (nirK)-like genes in marine thaumarchaea. *The ISME journal*, 6(10), 1966-1977.

Norton, J. M., Low, J. M., & Klotz, M. G. (1996). The gene encoding ammonia monooxygenase subunit A exists in three nearly identical copies in *Nitrosospira* sp. NpAV. *FEMS Microbiology Letters*, 139(2-3), 181-188.

Stein, L. Y., Arp, D. J., Berube, P. M., Chain, P. S., Hauser, L., Jetten, M. S., ... & Wei, X. (2007). Whole-genome analysis of the ammonia-oxidizing bacterium, *Nitrosomonas eutropha* C91: implications for niche adaptation. *Environmental microbiology*, 9(12), 2993-3007.

L. 79; Replace the first "explained" in this sentence

Response: Fixed.

L. 147; First use of formula NO₃  use nitrate (NO₃) instead

Response: We have now used nitrate (NO₃⁻) on line 143.

L.225-227; I'm still not entire sure what the difference is between these two correlations of amoA with temperature (other than one being linear and one exponential)

Response: Our apologies, these were redundant sentences. We have now removed the first sentence (lines 215-217).

L. 257-259; This final sentence remains a bit vague. What kind of changes would be referred to? How would this study be a foundation. I am not sure if such a generic final sentence is really necessary. The previous sentence may be enough as a closing statement. Otherwise, please be more specific.

Response: We removed the last sentence as suggested by the reviewer.

REVIEWERS' COMMENTS